# Endothelial pannexin 1–TRPV4 channel signaling lowers pulmonary arterial pressure in mice

**Zdravka Daneva[1], Matteo Ottolini[1,2], Yen Lin Chen[1], Eliska Klimentova[1], Maniselvan Kuppusamy[1], Soham A Shah[3], Richard D Minshall[4], Cheikh I Seye[5], Victor E Laubach[6], Brant E Isakson[7], Swapnil K Sonkusare[1,7]\***

[1]Robert M. Berne Cardiovascular Research Center, University of Virginia, Charlottesville, United States; [2]Department of Pharmacology, University of Virginia, Charlottesville, United States; [3]Department of Biomedical Engineering, University of Virginia, Charlottesville, United States; [4]Department of Anesthesiology, Department of Pharmacology, University of Illinois, Chicago, United States; [5]Department of Biochemistry, University of Missouri-Columbia, Columbia, United States; [6]Department of Surgery, University of Virginia, Charlottesville, United States; [7]Department of Molecular Physiology and Biological Physics, University of Virginia, Charlottesville, United States

**Abstract** Pannexin 1 (Panx1), an ATP-efflux pathway, has been linked with inflammation in pulmonary capillaries. However, the physiological roles of endothelial Panx1 in the pulmonary vasculature are unknown. Endothelial transient receptor potential vanilloid 4 (TRPV4) channels lower pulmonary artery (PA) contractility and exogenous ATP activates endothelial TRPV4 channels. We hypothesized that endothelial Panx1–ATP–TRPV4 channel signaling promotes vasodilation and lowers pulmonary arterial pressure (PAP). Endothelial, but not smooth muscle, knockout of Panx1 increased PA contractility and raised PAP in mice. Flow/shear stress increased ATP efflux through endothelial Panx1 in PAs. Panx1-effluxed extracellular ATP signaled through purinergic P2Y2 receptor (P2Y2R) to activate protein kinase Cα (PKCα), which in turn activated endothelial TRPV4 channels. Finally, caveolin-1 provided a signaling scaffold for endothelial Panx1, P2Y2R, PKCα, and TRPV4 channels in PAs, promoting their spatial proximity and enabling signaling interactions. These results indicate that endothelial Panx1–P2Y2R–TRPV4 channel signaling, facilitated by caveolin-1, reduces PA contractility and lowers PAP in mice.

**\*For correspondence:**
swapnil.sonkusare@virginia.edu

**Competing interest:** The authors declare that no competing interests exist.

## Introduction

The pulmonary endothelium exerts a dilatory influence on small, resistance-sized pulmonary arteries (PAs) and thereby lowers pulmonary arterial pressure (PAP). However, endothelial signaling mechanisms that control PA contractility remain poorly understood. In this regard, pannexin 1 (Panx1), which is expressed in the pulmonary endothelium and epithelium (*Navis et al., 2020*), has emerged as a crucial controller of endothelial function (*Begandt et al., 2017*; *Good et al., 2015*). Panx1, the most studied member of the pannexin family, forms a hexameric transmembrane channel at the cell membrane that allows efflux of ATP from the cytosol (*Bao et al., 2004*; *Lohman et al., 2012*). Previous studies indicated that flow/shear stress increases ATP efflux through Panx1 in EC monolayers (*Wang et al., 2016*). Endothelial Panx1 (Panx1$_{EC}$) has also been linked to inflammation in pulmonary capillaries (*Sharma et al., 2018*). Beyond this, however, the physiological roles of Panx1$_{EC}$ in the pulmonary vasculature are largely unknown.

Recent studies show that endothelial transient receptor potential vanilloid 4 (TRPV4$_{EC}$) channels reduce PA contractility and lower resting PAP (*Daneva et al., 2021*). Ca$^{2+}$ influx through TRPV4$_{EC}$ channels activates endothelial nitric oxide synthase (eNOS; *Marziano et al., 2017*) to dilate PAs. Moreover, exogenous ATP increases the activity of TRPV4$_{EC}$ channels in PAs (*Marziano et al., 2017*), although the regulation of TRPV4$_{EC}$ channels by endogenously released ATP is not known. We postulated that Panx1$_{EC}$-effluxed ATP acts through TRPV4$_{EC}$ channels to reduce PA contractility and lower PAP.

Purinergic receptor signaling is an essential regulator of pulmonary vascular function (*Lyubchenko et al., 2011*; *McMillan et al., 1999*; *Yamamoto et al., 2003*; *Konduri and Mital, 2000*). Extracellular ATP (eATP) is an endogenous activator of purinergic receptor signaling. However, the purinergic receptor subtype involved in eATP-induced activation of TRPV4$_{EC}$ channels has not been identified (*Marziano et al., 2017*). The pulmonary endothelium expresses both P2Y and P2X receptor subtypes. Konduri et al. showed that eATP dilates PAs through P2Y2 receptor (P2Y2R) activation and subsequent endothelial NO release (*Konduri and Mital, 2000*). These findings raise the possibility that the endothelial P2Y2 receptor (P2Y2R$_{EC}$) may be the signaling intermediate for Panx1$_{EC}$–TRPV4$_{EC}$ channel communication in PAs. The physiological roles of P2Y2R$_{EC}$ in the pulmonary vasculature remain unknown, mostly due to the lack of studies in PAs from endothelium-specific *P2ry2* conditional knockout mice (*P2ry2* cKO in EC).

The linkage between Panx1$_{EC}$-mediated eATP release and subsequent activation of P2Y2R$_{EC}$–TRPV4$_{EC}$ signaling could depend on the spatial proximity of individual elements—Panx1$_{EC}$, P2Y2R$_{EC}$, and TRPV4$_{EC}$—a functionality possibly provided by a signaling scaffold. Caveolin-1 (Cav-1) is a structural protein that interacts with and stabilizes other proteins in the pulmonary circulation (*Bernatchez et al., 2005*). Notably, endothelium-specific *Cav1* conditional knockout (*Cav1* cKO-EC) mice showed reduced TRPV4$_{EC}$ channel activity and elevated resting PAP (*Daneva et al., 2021*), supporting a crucial role for Cav-1 in TRPV4$_{EC}$ regulation of PAP. Although Cav-1 has also been shown to co-localize with Panx1 and P2Y2R in other cell types (*Goedicke-Fritz et al., 2015*; *DeLalio et al., 2018*; *Martinez et al., 2016*), its role in endothelial Panx1–P2Y2R signaling is not known.

Here, we tested the hypothesis that Panx1$_{EC}$–P2Y2R$_{EC}$–TRPV4$_{EC}$ channel signaling, supported by a signaling scaffold provided by Cav-1$_{EC}$, reduces PA contractility and PAP. Using inducible, EC-specific *Panx1*, *Trpv4*, *P2ry2*, and *Cav1* cKO mice, we show that endothelial Panx1–P2Y2R–TRPV4 signaling reduces PA contractility and lowers PAP. Panx1$_{EC}$-generated eATP acts via P2Y2R$_{EC}$ stimulation to activate protein kinase Cα (PKCα) and thereby increase TRPV4$_{EC}$ channel activity. Flow/shear stress is the physiological activator of ATP efflux through Panx1$_{EC}$ in PAs. Panx1$_{EC}$, P2Y2R$_{EC}$, PKCα, and TRPV4$_{EC}$ channels co-localize with Cav-1$_{EC,}$ ensuring spatial proximity among the individual elements and supporting signaling interactions. Overall, these findings advance our understanding of endothelial mechanisms that control PAP and suggest the possibility of targeting these mechanisms to lower PAP in pulmonary vascular disorders.

## Results

### Endothelial Panx1-mediated ATP release activates TRPV4$_{EC}$ signaling

The regulation of TRPV4$_{EC}$ channels by endogenously released ATP remains unknown. We postulated that ATP efflux through endothelial Panx1 promotes TRPV4$_{EC}$ channel activity. First, we determined the effect of eATP-hydrolyzing enzyme, apyrase (10 U/mL), on TRPV4$_{EC}$ channel activity in PAs from tamoxifen-inducible, EC-specific *Panx1* conditional knockout (*Panx1* cKO-EC) mice (*Lohman et al., 2015*) and tamoxifen-injected *Panx1*$^{fl/fl}$ Cre$^-$ (*Panx1*$^{fl/fl}$) control mice (*Figure 1A*, *Figure 1—figure supplement 1*; *Sharma et al., 2018*). *En face* PAs from *Panx1* cKO-EC mice displayed a lack of endothelial (CD31, green) Panx1 immunostaining (red). Localized, unitary Ca$^{2+}$ influx signals through TRPV4$_{EC}$ channels, termed TRPV4$_{EC}$ sparklets (*Sonkusare et al., 2012*), were recorded in *en face*, fourth-order PAs (~50 μm) loaded with Fluo-4. Addition of apyrase reduced the activity of TRPV4$_{EC}$ sparklets in PAs from control mice, confirming the regulation of TRPV4$_{EC}$ channels by endogenous eATP (*Figure 1A*). However, apyrase was unable to decrease TRPV4$_{EC}$ sparklet activity in PAs from *Panx1* cKO-EC mice, suggesting that endothelial Panx1 may be a critical source of eATP in PAs (*Figure 1A*).

Bioluminescence measurements confirmed lower baseline eATP levels in PAs from *Panx1* cKO-EC mice than PAs from *Panx1*$^{fl/fl}$ control mice (*Figure 1B*), supporting an essential role for Panx1$_{EC}$ as an eATP-release mechanism in PAs. PAs from *Trpv4* cKO-EC mice, however, exhibited unaltered basal

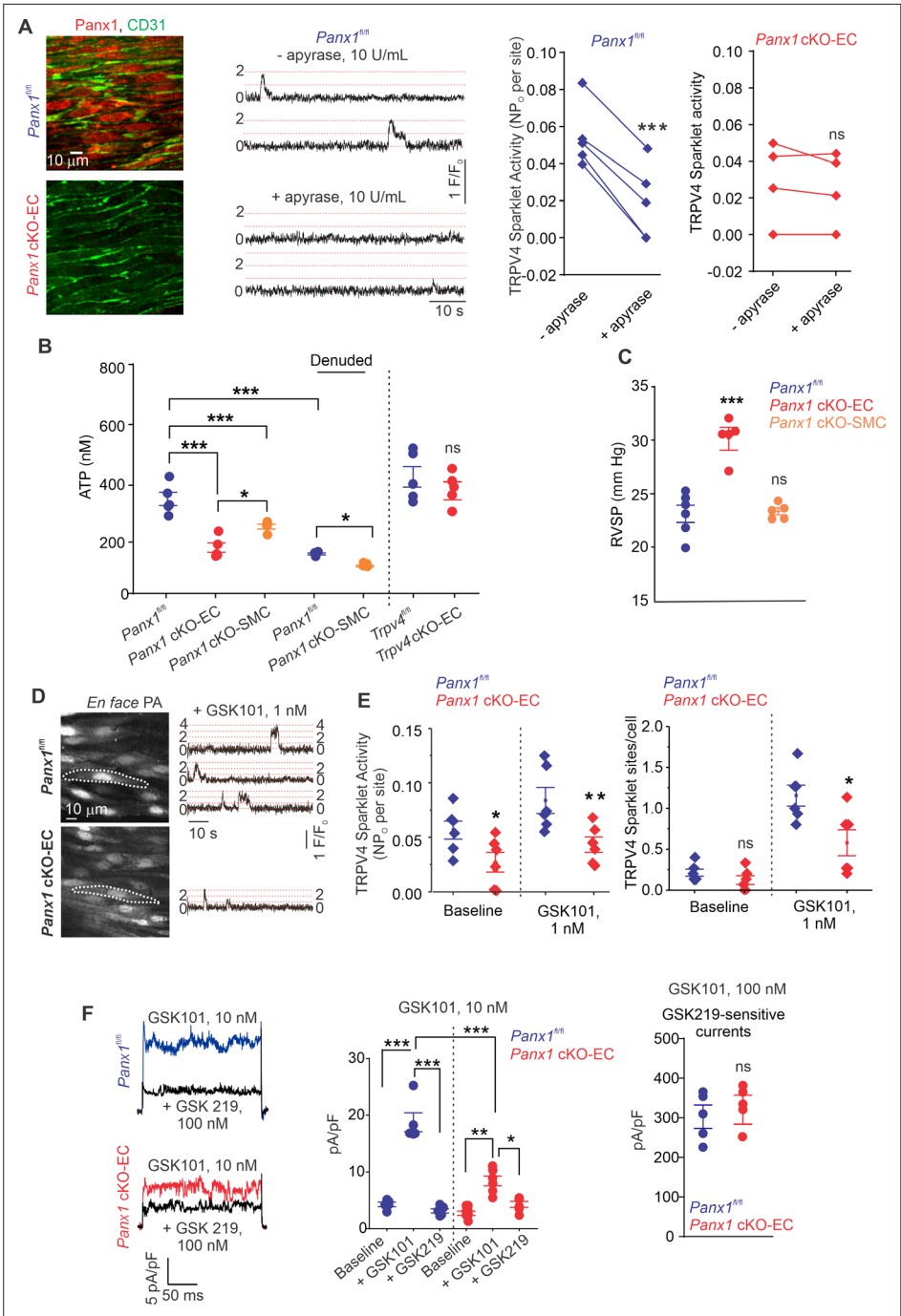

**Figure 1.** ATP efflux through Panx1_EC ATP activates TRPV4_EC channels in pulmonary arteries (PAs) and lowers pulmonary arterial pressure (PAP). (**A**) Left: immunofluorescence images of *en face* fourth-order PAs from *Panx1*[fl/fl] and *Panx1* cKO-EC mice. CD31 immunofluorescence indicates ECs. Center: representative traces showing TRPV4_EC sparklet activity in *en face* preparations of PAs from *Panx1*[fl/fl] mice in the absence or presence of apyrase (10 U/mL). Dotted lines are quantal levels. Experiments were performed in Fluo-4-loaded PAs in the presence of cyclopiazonic acid (CPA; 20 μmol/L CPA, included to eliminate $Ca^{2+}$ release from intracellular stores). Right: TRPV4_EC sparklet activity ($NP_o$) per site in *en face* preparations of PAs from *Panx1*[fl/fl] and *Panx1* cKO-EC mice in the presence or absence of apyrase (10 U/mL; n = 5; ***p<0.001 vs. *Panx1*[fl/fl] [-apyrase, 10 U/mL]; ns indicates no statistical significance; t-test). 'N' is the number of channels per site and '$P_O$' is the open state probability of the channel. (**B**), measurements of ATP (nmol/L) levels in PAs from *Panx1*[fl/fl], *Panx1* cKO-EC, *Panx1* cKO-SMC, *Trpv4*[fl/fl], and *Trpv4* cKO-EC mice, and endothelium-denuded PAs from *Panx1*[fl/fl] and *Panx1* cKO-SMC mice (n = 5–6; *p<0.05 vs. *Panx1* cKO-EC; *p<0.05 vs. *Panx1*[fl/fl] [denuded]; ***p<0.001 vs. *Panx1*[fl/fl]; ***p<0.001 vs. *Panx1* cKO-SMC; ns indicates

*Figure 1 continued on next page*

*Figure 1 continued*

no statistical significance; one-way ANOVA). (**C**) Average resting right ventricular systolic pressure (RVSP) values in *Panx1*$^{fl/fl}$, *Panx1* cKO-EC, and *Panx1* cKO-SMC mice (n = 6; ***p<0.001 vs. *Panx1*$^{fl/fl}$; ns indicates no statistical significance; one-way ANOVA). (**D**) Left grayscale image of a field of view in an *en face* preparation of Fluo-4-loaded PAs from *Panx1*$^{fl/fl}$ and *Panx1* cKO-EC mice showing approximately 20 ECs. Dotted outlines indicate an EC (20 µmol/L CPA included to eliminate Ca$^{2+}$ release from intracellular stores). Right: representative traces showing TRPV4$_{EC}$ sparklet activity in *en face* preparations of PAs from *Panx1*$^{fl/fl}$ and *Panx1* cKO-EC mice in response to GSK1016790A (GSK101; 1 nmol/L). Experiments were performed in Fluo-4-loaded PAs in the presence of CPA (20 µmol/L). (**E**) TRPV4$_{EC}$ sparklet activity (NP$_O$) per site and sites per cell in *en face* preparations of PAs from *Panx1*$^{fl/fl}$ and *Panx1* cKO-EC mice under baseline conditions (i.e., 20 µmol/L CPA) and in response to 1 nmol/L GSK101 (n = 6; *p<0.05, **p<0.01 vs. *Panx1*$^{fl/fl}$; *p<0.05 vs. *Panx1*$^{fl/fl}$; ns indicates no statistical significance; two-way ANOVA). (**F**) Left: representative GSK101 (10 nmol/L)-induced outward TRPV4$_{EC}$ currents in freshly isolated ECs from *Panx1*$^{fl/fl}$ and *Panx1* cKO-EC mice and effect of GSK2193874 (GSK219, TRPV4 inhibitor, 100 nmol/L) in the presence of GSK101. Currents were elicited by a 200 ms voltage step from –50 mV to +100 mV. Center: scatterplot showing outward currents at +100 mV under baseline conditions, after the addition of GSK101 (10 nmol/L), and after the addition of GSK219 (100 nmol/L; n = 5–6 cells, *p<0.05 vs. *Panx1* cKO-EC [+GSK101]; **p<0.01 vs. *Panx1* cKO-EC [baseline]; ***p<0.001 vs. *Panx1*$^{fl/fl}$ [+baseline]; vs. *Panx1*$^{fl/fl}$ [+GSK101]; and *Panx1* cKO-EC [+GSK101] vs. *Panx1*$^{fl/fl}$ [+GSK101]; two-way ANOVA). Right: scatterplot showing GSK219-sensitive TRPV4$_{EC}$ currents in response to GSK101 (100 nmol/L; ns indicates no statistical significance; n = 5).

The online version of this article includes the following figure supplement(s) for figure 1:

**Figure supplement 1.** Panx1$_{SMC}$ mRNA levels in mesenteric arteries from *Panx1*$^{fl/fl}$ and *Panx1* cKO-SMC mice.

**Figure supplement 2.** TRPV4$_{EC}$ sparklet activity (NP$_O$) per site and TRPV4 sparklet sites per cell in *en face* preparations of pulmonary arteries (PAs) from *Panx1*$^{fl/fl}$ and *Panx1* cKO-EC mice in response to 30 nmol/L GSK101.

eATP levels, suggesting that TRPV4$_{EC}$ channels do not regulate Panx1$_{EC}$ activity under basal conditions. Although eATP levels were also reduced in PAs from inducible, smooth muscle cell-specific *Panx1* cKO (*Panx1* cKO-SMC) (*Good et al., 2018*) mice, the eATP levels in these mice were higher than *Panx1* cKO-EC mice (*Figure 1B*, *Figure 1—figure supplement 1*). Endothelial denudation also reduced eATP levels in PAs from control mice, which were reduced further in endothelium-denuded PAs from *Panx1* cKO-SMC mice.

We recently demonstrated that right ventricular systolic pressure (RVSP), a commonly used in vivo indicator of PAP, was elevated in inducible EC-specific *Trpv4* KO (*Trpv4* cKO-EC) mice (*Daneva et al., 2021*). Similarly, *Panx1* cKO-EC mice also showed elevated RVSP (*Figure 1C*). The Fulton index, a ratio of right ventricular (RV) weight to left ventricle plus septal (LV + S) weight, was not altered in *Panx1* cKO-EC mice compared to control mice, suggesting a lack of right ventricular hypertrophy in these mice (*Table 1*). Baseline RVSP was not altered in *Panx1* cKO-SMC mice (*Figure 1C*), indicating a lack of regulation of resting PAP by SMC Panx1. Functional cardiac MRI studies indicated no alterations in cardiac function in *Panx1* cKO-EC mice compared to the control mice (*Table 1*), confirming that the changes in RVSP were not due to altered cardiac function.

Baseline TRPV4$_{EC}$ sparklet activity and that induced by a low concentration (1 nmol/L) of the specific TRPV4 channel agonist, GSK1016790A (hereafter, GSK101), were significantly reduced in PAs from *Panx1* cKO-EC mice compared to PAs from *Panx1*$^{fl/fl}$ mice (*Figure 1D and E*). Additionally, the number of TRPV4$_{EC}$ sparklet sites per cell was decreased in PAs from *Panx1* cKO-EC mice (*Figure 1E*). At the agonist concentration that maximally activates TRPV4$_{EC}$ sparklets in PAs (30 nmol/L GSK101; *Daneva et al., 2021*), sparklet activity per site and sparklet sites per cell were not different between *Panx1* cKO-EC Panx1 and control mice (*Figure 1—figure supplement*

**Table 1.** Fulton index and functional MRI analysis of cardiac function in *Panx1*$^{fl/fl}$ and *Panx1* cKO-EC mice.

Average Fulton index, end diastolic and systolic volume (EDV and ESV; µL), ejection fraction (EF; %), stroke volume (SV; µL), R-R interval (ms), and cardiac output (CO; mL/min). Data are presented as means ± SEM (n = 5–8 mice).

|  | *Panx1*$^{fl/fl}$ | *Panx1* cKO-EC |
|---|---|---|
| Fulton index | 0.23 ± 0.01 | 0.26 ± 0.03 |
| EDV (µL) | 46.9 ± 2.7 | 50.9 ± 2.9 |
| ESV (µL) | 14.8 ± 1.7 | 13.1 ± 1.4 |
| EF (%) | 68.9 ± 2.0 | 74.3 ± 2.3 |
| SV (µL) | 32.2 ± 1.3 | 37.8 ± 2.4 |
| R-R (ms) | 127.1 ± 5.5 | 130.8 ± 2.5 |
| CO (mL/min) | 15.2 ± 0.6 | 17.3 ± 1.2 |

2). Outward currents through TRPV4$_{EC}$ channels, elicited by 10 nmol/L GSK101, were also lower in *Panx1* cKO-EC than *Panx1*$^{fl/fl}$ mice (*Figure 1F*, left and center). However, when maximally activated, TRPV4$_{EC}$ channel currents were not different between *Panx1* cKO-EC and *Panx1*$^{fl/fl}$ mice (*Figure 1F*, right), suggesting that the maximum number of functional TRPV4$_{EC}$ channels is not altered in *Panx1* cKO-EC mice.

## Endothelial Panx1–TRPV4 signaling lowers pressure- and agonist-induced PA constriction

Isolated, pressurized PAs (50–100 µm, *Figure 2A*) from *Trpv4* cKO-EC mice exhibited a greater intraluminal pressure-induced (myogenic) constriction than PAs from control mice (*Figure 2B*, *Figure 2—figure supplement 1*), providing the first evidence that TRPV4$_{EC}$ channels oppose myogenic constriction in PAs. This finding was further supported by a greater contractile response to the thromboxane A$_2$ receptor agonist U46619 in PAs from *Trpv4* cKO-EC mice (1–300 nmol/L; *Figure 2C*). PAs from *Panx1* cKO-EC mice also showed a higher myogenic constriction than PAs from control mice (*Figure 2D*), offering the first evidence that endothelial Panx1 regulates myogenic constriction of PAs. U46619-induced constriction was also increased in PAs from *Panx1* cKO-EC mice compared to PAs from control mice. Pretreatment of PAs from *Panx1* cKO-EC mice with a low concentration of TRPV4 agonist (GSK101, 3 nmol/L) reduced the U46619-induced constriction to control levels, indicating that endothelial Panx1 dilates PAs through TRPV4$_{EC}$ channels. The presence of apyrase also increased U46619-induced constriction of PAs from control mice, confirming the dilatory effect of eATP on PAs (*Figure 2—figure supplement 2*). Further, exogenous ATP-induced dilation was absent in PAs from *Trpv4* cKO-EC mice (*Figure 2—figure supplement 3*, center) but was not affected in PAs from *Panx1* cKO-EC mice (*Figure 2—figure supplement 3*, right), supporting the concept that ATP-TRPV4$_{EC}$ channel signaling occurs downstream of Panx1$_{EC}$. Together, these data provide the first evidence that Panx1$_{EC}$–eATP–TRPV4$_{EC}$ channel signaling lowers PA contractility and resting PAP.

To verify the possibility that flow/shear stress activates ATP efflux through endothelial Panx1, we measured luminal eATP levels in PAs following exposure to different intraluminal shear stress levels (4, 7, and 14 dynes/cm$^2$; *Figure 2F*; *Ahn et al., 2017*). Increase in shear stress elevated luminal eATP levels in PAs from control mice, but not in PAs from *Panx1* cKO-EC mice (*Figure 2G*), confirming a critical role for Panx1$_{EC}$ in shear stress-induced increase in luminal eATP. Also, shear stress-induced increase in luminal eATP was not altered in PAs from *Trpv4* cKO-EC mice compared to control mice (*Figure 2H*), suggesting that TRPV4$_{EC}$ channels do not influence the efflux of ATP through Panx1$_{EC}$ in response to increase in shear stress. eATP acts through purinergic P2Y2R$_{EC}$ stimulation to activate TRPV4$_{EC}$ channels.

The main P2Y receptor subtypes in the pulmonary endothelium are P2Y1R and P2Y2R (*Konduri and Mital, 2000*; *Konduri et al., 2004*; *Zemskov et al., 2011*). The selective P2Y1R inhibitor MRS2179 (MRS, 10 µmol/L) did not alter eATP activation of TRPV4$_{EC}$ sparklets (*Figure 3A*). In contrast, the selective P2Y2R inhibitor AR-C 118925XX  (AR-C; 10 µmol/L) completely abrogated the effect of eATP on TRPV4$_{EC}$ sparklets (*Figure 3A*). eATP was also unable to activate TRPV4$_{EC}$ sparklets in inducible, endothelium-specific EC-specific *P2ry2* cKO-EC mice (*Figure 3A*), providing further evidence that eATP activates TRPV4$_{EC}$ channels in PAs specifically via P2Y2R$_{EC}$ signaling. The general P2X1-5 receptor inhibitor, PPADS (10 µmol/L), and P2X7 receptor inhibitor, JNJ-47965567 (JNJ, 1 µmol/L), did not alter the effect of eATP on TRPV4$_{EC}$ sparklets, ruling out a role for P2X1-5/7 receptors in eATP activation of TRPV4$_{EC}$ channels in PAs (*Figure 3B*). In ECs freshly isolated from PAs of C57BL6 mice, ATP (10 µmol/L) increased the outward currents through TRPV4$_{EC}$ channels (*Figure 3C*). Furthermore, the selective P2Y2R agonist, 2-thiouridine-5′-triphosphate (2-thio UTP; 0.5 µmol/L) activated TRPV4$_{EC}$ sparklets in PAs from *P2ry2*$^{fl/fl}$ mice but not in PAs from *P2ry2* cKO-EC mice (*Figure 3D*).

Similar to *Panx1* cKO-EC mice, *P2ry2* cKO-EC mice showed elevated RVSP and unaltered Fulton index (*Figure 3E*). Exogenous ATP (1 µmol/L)-induced dilation was abolished in PAs from *P2ry2* cKO-EC mice (*Figure 3F*), confirming an essential role of P2Y2R$_{EC}$ in ATP-induced dilation of PAs. Further, PAs from *P2ry2* cKO-EC mice showed higher myogenic and U46619-induced constriction compared to PAs from control mice (*Figure 3G*). As observed with PAs from *Panx1* cKO-EC mice, pretreatment with a low concentration of TRPV4 channel agonist (GSK101, 3 nmol/L) reduced U46619-induced constriction to control levels in PAs from *P2ry2* cKO-EC mice (*Figure 3H*). Taken together, these findings

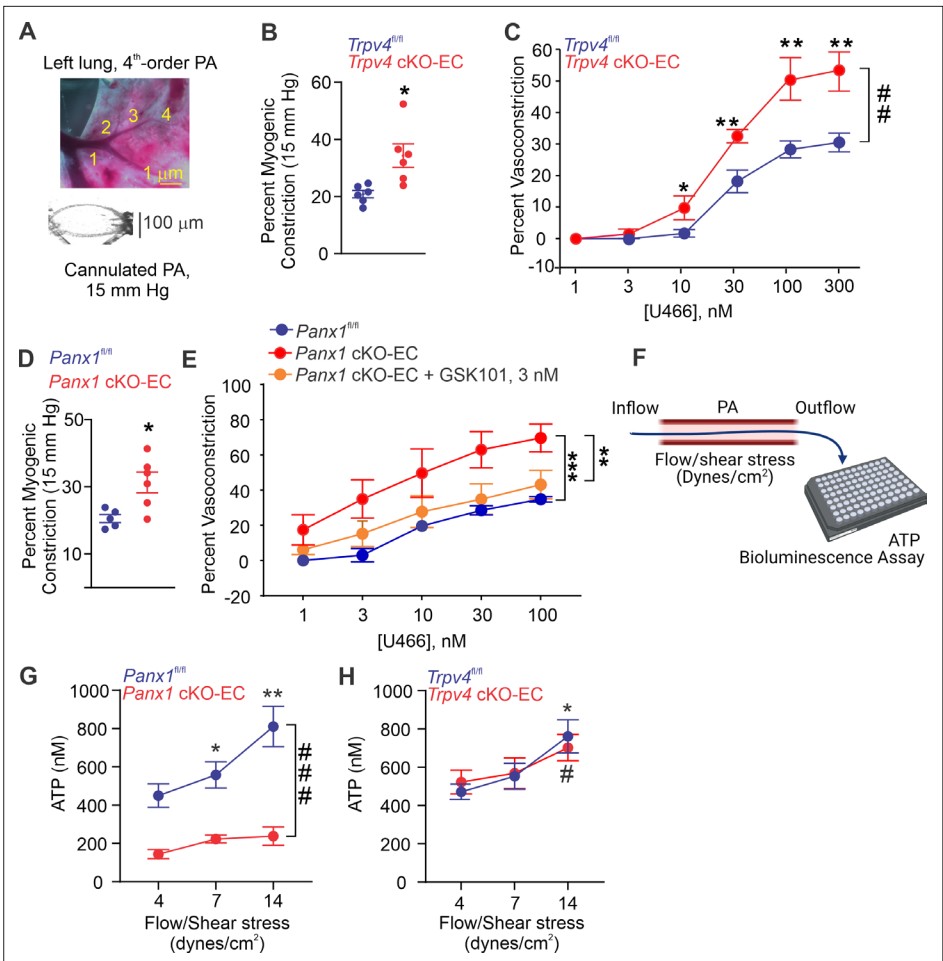

**Figure 2.** Endothelial Panx1–TRPV4 signaling lowers myogenic and agonist-induced constriction of pulmonary arteries (PAs). (**A**) Top: an image showing the left lung and the order system used to isolate fourth-order PAs in this study; bottom: an image of a fourth-order PA cannulated and pressurized at 15 mm Hg. (**B**) Percentage myogenic constriction of PAs from $Trpv4^{fl/fl}$ and $Trpv4$ cKO-EC mice (n = 6; *p<0.05; t-test). (**C**) Percent constriction of PAs from $Trpv4^{fl/fl}$ and $Trpv4$ cKO-EC mice in response to thromboxane A2 receptor agonist U46619 (U466, 1–300 nmol/L; n = 5; *p<0.05 vs. $Trpv4^{fl/fl}$ [10 nmol/L], **p<0.01 vs. $Trpv4^{fl/fl}$ [30, 100, and 300 nmol/L]; ##p<0.01 vs. $Trpv4^{fl/fl}$; two-way ANOVA). (**D**) Percentage myogenic constriction of PAs from $Panx1^{fl/fl}$ and $Panx1$ cKO-EC mice (n = 6; *p<0.05; t-test). (**E**) U46619 (U466, 1–300 nmol/L)-induced constriction of PAs from $Panx1^{fl/fl}$, $Panx1$ cKO-EC, and $Panx1$ cKO-EC mice in the absence or presence of GSK101 (3 nmol/L) (n = 5; **p<0.01 vs. $Panx1$ cKO-EC, ***p<0.01 vs. $Panx1^{fl/fl}$; two-way ANOVA, between groups). (**F**) Schematic of flow-induced ATP release from isolated and cannulated fourth-order PAs. Shear stress was calculated using the following equation: $\tau = 4(\mu\dot{Q})/(\pi r^3)$, where μ is viscosity, $\dot{Q}$ is volumetric flow, and $r$ is internal radius of the vessel. Outflow was collected every 10 min and ATP was measured using Luciferin-Luciferase ATP Bioluminescence Assay. (**G**) Release of ATP (nmol/L) from PAs of $Panx1^{fl/fl}$ and $Panx1$ cKO-EC mice in response to flow/shear stress in the presence of ARL-67156 (ARL; ecto-ATPase inhibitor; 300 μmol/L; 4, 7, and 14 dynes/cm²; n = 6; *p<0.05 vs. $Panx1^{fl/fl}$ [4 dynes/cm²]; **p<0.01 vs. $Panx1^{fl/fl}$ [7 dynes/cm²]; ###p<0.001 vs. $Panx1$ cKO-EC; two-way ANOVA). (**H**) Release of ATP (nmol/L) from PAs of $Trpv4^{fl/fl}$ and $Trpv4$ cKO-EC mice in response to flow/shear stress in the presence of ARL (300 μmol/L; 4, 7, and 14 dynes/cm²; n = 6; *p<0.05 vs. $Trpv4^{fl/fl}$ [4 dynes/cm²]; #p<0.05 vs. $Trpv4$ cKO-EC [4 dynes/cm²]; two-way ANOVA).

The online version of this article includes the following source data and figure supplement(s) for figure 2:

**Source data 1.** Endothelial TRPV4 knockout increases U46619-induced constriction of PAs.

**Source data 2.** Endothelial Panx1 knockout increases U46619-induced constriction of PAs.

**Source data 3.** Shear stress increases ATP efflux through endothelial Panx1 in PAs.

**Source data 4.** Endothelial TRPV4 channel does not contribute to shear stress-induced increase in luminal ATP.

**Figure supplement 1.** Percent myogenic constriction in small pulmonary arteries (PAs; 50–100 μm internal

*Figure 2 continued on next page*

*Figure 2 continued*

diameter) and large PAs (>200 μm internal diameter; n = 6–10; ***p<0.001).

**Figure supplement 2.** Percent constriction of pulmonary arteries (PAs) from *Panx1*^fl/fl^ and *Panx1*^fl/fl^ plus apyrase (10 U/mL) mice in response to U46619 (U466; 1–100 nmol/L; n = 5; **p<0.01 vs. *Panx1*^fl/fl^; two-way ANOVA).

**Figure supplement 2—source data 1.** Apyrase increases U46619-induced constriction of PAs.

**Figure supplement 3.** Left: representative diameter traces showing ATP (1 μmol/L)-induced dilation of pulmonary arteries (PAs) from *Trpv4*^fl/fl^ and *Trpv4* cKO-EC mice, pre-constricted with the thromboxane A2 receptor analog U46619 (50 nmol/L).

demonstrate that P2Y2R$_{EC}$ is the signaling intermediate for Panx1$_{EC}$–TRPV4$_{EC}$ channel interaction in PAs.

## Cav-1$_{EC}$ provides a scaffold for Panx1$_{EC}$–P2Y2R$_{EC}$–TRPV4$_{EC}$ signaling

We hypothesized that Cav-1$_{EC}$ provides a signaling scaffold that supports and maintains the spatial proximity among the individual elements in the Panx1$_{EC}$–P2Y2R$_{EC}$–TRPV4$_{EC}$ pathway. Previous studies demonstrated that endothelium-specific knockout of *Cav1* results in reduced TRPV4$_{EC}$ channel current density and elevated PAP (*Daneva et al., 2021*). Here, we provide evidence that eATP-induced activation of TRPV4$_{EC}$ sparklets is absent in PAs from *Cav1* cKO-EC mice (*Figure 4A*; knockout validation in *Daneva et al., 2021*). As observed with PAs from *Trpv4* cKO-EC and *P2ry2* cKO-EC mice, eATP-induced dilation was also abolished in PAs from *Cav1* cKO-EC mice (*Figure 4B*). These results provided the first functional evidence that Cav-1$_{EC}$ is required for eATP–P2Y2R$_{EC}$–TRPV4$_{EC}$ signaling in PAs. To provide additional evidence to support Cav-1$_{EC}$–dependent co-localization of Panx1$_{EC}$–P2Y2R$_{EC}$–TRPV4$_{EC}$ signaling elements in PAs, we performed in situ proximity ligation assays (PLAs), which allow the detection of two proteins in close proximity (<40 nm). PLA data confirmed that Cav-1$_{EC}$ exists within nanometer proximity of Panx1$_{EC}$, P2Y2R$_{EC}$, and TRPV4$_{EC}$ channels in PAs (*Figure 4C*). Nanometer proximity was also observed between TRPV4$_{EC}$ channels and P2Y2R$_{EC}$ and between Panx1$_{EC}$ and P2Y2R$_{EC}$ (*Figure 4D*, *Figure 4—figure supplement 1*). TRPV4$_{EC}$:P2Y2R and P2Y2R:Panx1 co-localization was lost in PAs from *Cav1* cKO-EC mice, further supporting the crucial scaffolding role of Cav-1$_{EC}$ in Panx1$_{EC}$–P2Y2R$_{EC}$–TRPV4$_{EC}$ pathway. PA endothelium has also been shown to express another P2Y family receptor, P2Y1 (P2Y1R) (*Konduri et al., 2004*). The PLA data confirmed that P2Y1R does not occur in nanometer proximity with Cav-1$_{EC}$ in PAs (*Figure 4—figure supplement 2*). Together, these data confirmed a crucial role for Cav-1$_{EC}$ in facilitating the spatial proximity amongst the individual elements of the Panx1$_{EC}$–P2Y2R$_{EC}$–TRPV4$_{EC}$ pathway.

## Cav-1$_{EC}$ anchoring of PKCα mediates P2Y2R$_{EC}$-dependent activation of TRPV4$_{EC}$ channels in PAs

P2Y2R is a Gq protein-coupled receptor that activates the phospholipase C (PLC)–diacylglycerol (DAG)–PKC signaling pathway. Notably, PKC is known to phosphorylate TRPV4 channels and potentiate its activity (*Fan et al., 2009*). eATP, the DAG analog OAG (1 μmol/L), and the PKC activator phorbol myristate acetate (PMA; 10 nmol/L) stimulated TRPV4$_{EC}$ sparklet activity in small PAs (*Figure 5A, B and C*). Inhibition of PLC with U73122 (3 μmol/L) abolished eATP activation of TRPV4$_{EC}$ sparklets, but not OAG- or PMA-induced activation of TRPV4$_{EC}$ sparklets. Moreover, the PKCα/β inhibitor Gö-6976 (1 μmol/L) prevented activation of TRPV4$_{EC}$ sparklets by ATP, OAG, and PMA (*Figure 5A, B and C*), supporting the concept that eATP activation of P2Y2R$_{EC}$ stimulates TRPV4$_{EC}$ channel activity via PLC–DAG–PKC signaling in PAs. TRPV4$_{EC}$ channel activation by PLC–DAG–PKC signaling was further supported by increased activity of TRPV4$_{EC}$ sparklets in PAs from *Cdh5*-optoα1 adrenergic receptor (*Cdh5*-optoα1AR) mouse, which expresses light-sensitive α1AR in endothelial cells (*Figure 5D*). When activated with light (~473 nm), Optoα1AR generates the secondary messengers IP3 and diacylglycerol (DAG) (*Airan et al., 2009*). Light activation resulted in increased activity of TRPV4$_{EC}$ sparklets (*Figure 5D*, *Figure 5—figure supplement 1*), an effect that was abolished by the PKCα/β inhibitor Gö-6976 (1 μmol/L) and in the presence of specific TRPV4 inhibitor GSK2193874 (hereafter GSK219; 100 nmol/L; *Figure 5—figure supplement 2*).

Since Cav-1 possesses a PKC-binding domain (*Mineo et al., 1998*) and exists in nanometer proximity with TRPV4$_{EC}$ channels and P2Y2R$_{EC}$, we tested the hypothesis that Cav-1$_{EC}$ anchoring of PKC

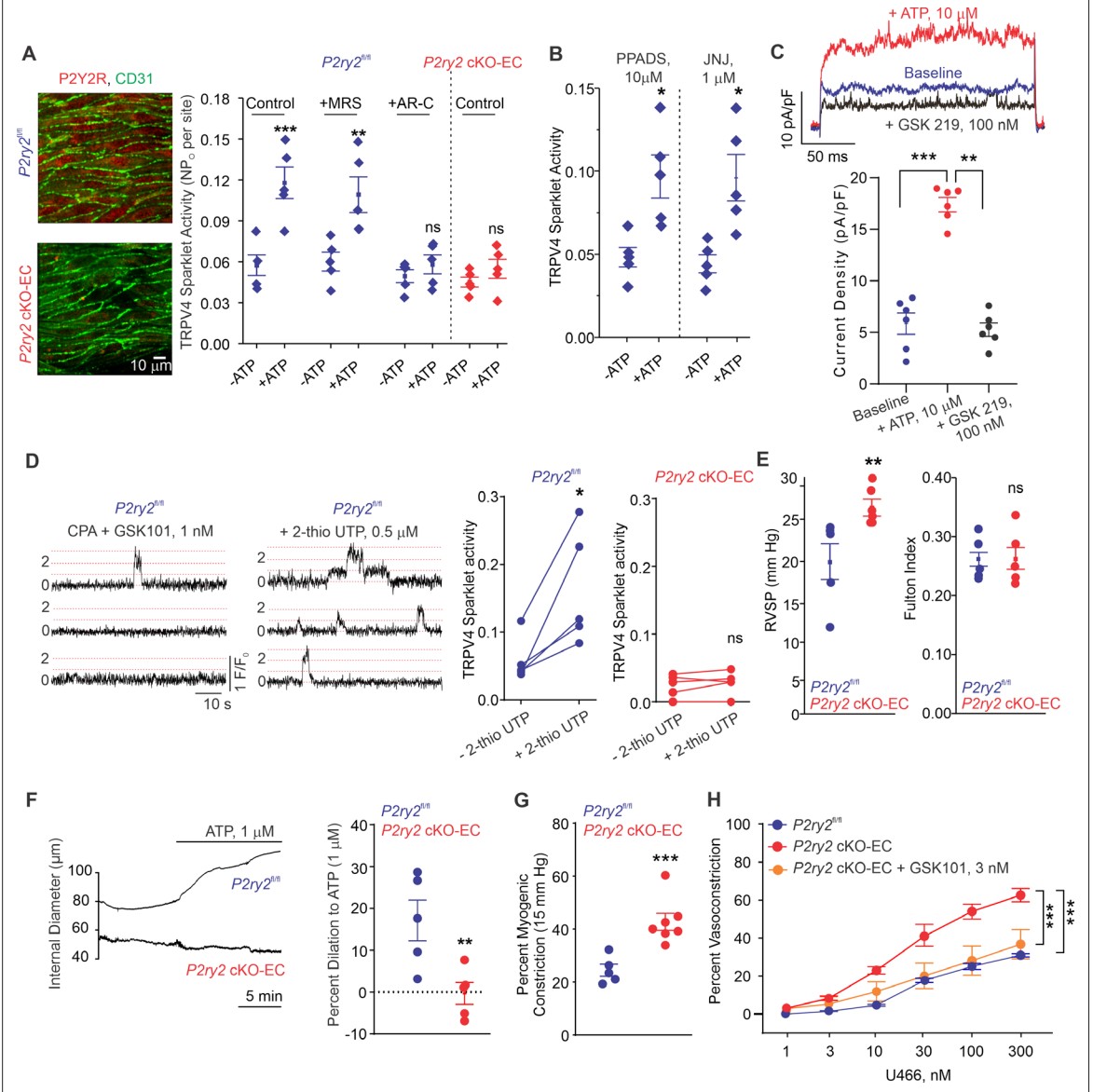

**Figure 3.** Endothelial P2Y2R-TRPV4 channel signaling lowers pulmonary artery (PA) contractility and pulmonary arterial pressure (PAP). (**A**) Left: immunofluorescence images of *en face* fourth-order PAs from *P2ry2*fl/fl and *P2ry2* cKO-EC mice. CD31 immunofluorescence indicates ECs. Right: effects of ATP (1 μmol/L) on TRPV4EC sparklet activity in the absence or presence of the P2Y1R inhibitor MRS2179 (MRS; 10 μmol/L) or P2Y2R inhibitor AR-C 118925XX  (AR-C; 10 μmol/L) in PAs from *P2ry2*fl/fl and *P2ry2* cKO-EC mice, expressed as NP$_O$ per site (n = 5; ***p<0.001 vs. Control [- ATP]; **p<0.01 vs.+ MRS [- ATP]; ns indicates no statistical significance; two-way ANOVA). 'N' is the number of channels per site and 'P$_O$' is the open state probability of the channel. (**B**) Effects of ATP (1 μmol/L) on TRPV4EC sparklet activity in the presence of the general P2X1-5/7R  inhibitor PPADS (10 μmol/L) and P2X7 R inhibitor JNJ-47965567 (JNJ; 1 μmol/L) in PAs of C57BL6/J mice (n = 5; *p<0.05 vs. [-ATP]; one-way ANOVA). (**C**) Top: representative ATP (10 μmol/L)-induced outward TRPV4 currents in freshly isolated ECs from C57BL6/J mice and the effect of GSK2193874 (GSK219; TRPV4 inhibitor; 100 nmol/L) in the presence of ATP. Currents were elicited by a 200 ms voltage step from –50 mV to +100 mV. Bottom: scatterplot showing outward currents at +100 mV under baseline conditions, after the addition of ATP, and after the addition of GSK219 (100 nmol/L; n = 6 cells; ***p<0.001 vs. baseline; **p<0.01 vs.+ ATP [10 μmol/L]; one-way ANOVA). (**D**) Left: representative traces showing TRPV4EC sparklet activity in *en face* preparations of PAs from *P2ry2*fl/fl mice. Dotted lines are quantal levels. Right: TRPV4EC sparklet activity per site (NP$_O$) in *en face* preparations of PAs from *P2ry2*fl/fl and *P2ry2* cKO-EC mice under baseline conditions (i.e., 20 μmol/L cyclopiazonic acid [CPA]) and in response to 2-thio UTP (P2Y2R agonist, 0.5 μmol/L; n = 5; *p<0.05 vs. *P2ry2*fl/fl [-2-thio UTP]; ns indicates no statistical significance; t-test). (**E**) Left: average resting right ventricular systolic pressure (RVSP) values in *P2ry2*fl/fl and *P2ry2* cKO-EC mice (n = 6; **p<0.01; t-test). Right: average Fulton index values in *P2ry2*fl/fl and *P2ry2* cKO-EC mice (n = 5–6; ns indicates no statistical significance). (**F**) Right: representative diameter traces showing ATP (1 μmol/L)-induced dilation of PAs from *P2ry2*fl/fl and *P2ry2* cKO-EC mice, pre-constricted with the thromboxane A2 receptor agonist U46619 (U466, 50 nmol/L). Fourth-order PAs were pressurized to 15 mm Hg. Right: percent dilation of PAs from *P2ry2*fl/fl and *P2ry2* cKO-EC mice in response to ATP (1 μmol/L; n = 5–10; ***p<0.01 vs. *P2ry2*fl/fl [ATP 1 μmol/L]; t-test). (**G**) Percentage myogenic constriction of

*Figure 3 continued on next page*

Figure 3 continued

PAs from *P2ry2*[fl/fl] and *P2ry2* cKO-EC mice (n = 5–7; ***p<0.001; t-test). (**H**) U46619 (U466, 1–300 nmol/L)-induced constriction of PAs from *P2ry2*[fl/fl], *P2ry2* cKO-EC, and *P2ry2* cKO-EC mice in the absence or presence of GSK101 (3 nmol/L) (n = 5; ***p<0.001 vs. *P2ry2* cKO-EC, ***p<0.001 vs. *P2ry2*[fl/fl]; two-way ANOVA).

The online version of this article includes the following source code for figure 3:

**Source data 1.** Endothelial P2Y2R knockout increases U46619-induced constriction of PAs.

mediates P2Y2R$_{EC}$–TRPV4$_{EC}$ channel interaction in PAs. PLA experiments confirmed that PKC also exists in nanometer proximity with Cav-1$_{EC}$ in PAs (**Figure 6A**). The PKC dependence of Cav-1$_{EC}$ activation of TRPV4$_{EC}$ channels was confirmed by studies in HEK293 cells transfected with TRPV4 alone or TRPV4 channels plus Cav-1 (**Figure 6B**), which showed that TRPV4 currents were increased in the presence of Cav-1. Further, the PKCα/β inhibitor Gö-6916 (1 µmol/L) reduced TRPV4 channel currents in Cav-1/ TRPV4-co-transfected cells to the level of that in cells transfected with TRPV4 alone (**Figure 6B and C**). These results imply that Cav-1 enhances TRPV4 channel activity via PKCα/β anchoring. Experiments in which TRPV4 channels were co-expressed with PKCα or PKCβ showed that only PKCα increased currents through TRPV4 channels (**Figure 6D**). Collectively, these results support the conclusion that Panx1$_{EC}$–P2Y2R$_{EC}$–PKCα–TRPV4$_{EC}$ signaling on a Cav-1$_{EC}$ scaffold reduces PA contractility and lowers resting PAP (**Figure 6E**).

## Discussion

Regulation of PA contractility and PAP is a complex process involving multiple cell types and signaling elements. In particular, the endothelial signaling mechanisms that control resting PAP remain poorly understood. Our studies identify a Panx1$_{EC}$, P2Y2R$_{EC}$, and TRPV4$_{EC}$ channel-containing signaling nanodomain that reduces PA contractility and lowers PAP. Although Panx1$_{EC}$ and P2Y2R$_{EC}$ have been implicated in the regulation of endothelial function, their impact on PAP remains unknown. We demonstrate critical roles for several key, linked mechanistic, pathways showing that (1) Panx1$_{EC}$ increases eATP levels in small PAs; (2) Panx1$_{EC}$-generated eATP, in turn, enhances Ca$^{2+}$ influx through TRPV4$_{EC}$ channels, thereby dilating PAs and lowering PAP; (3) eATP acts through purinergic P2Y2R$_{EC}$–PKCα signaling to activate TRPV4$_{EC}$ channels; and (4) Cav-1$_{EC}$ provides a signaling scaffold that ensures spatial proximity among the elements of the Panx1$_{EC}$–P2Y2R$_{EC}$–PKCα–TRPV4$_{EC}$ pathway. Our findings reveal a novel signaling axis that can be engaged by physiological stimuli to lower PAP and could also be therapeutically targeted in pulmonary vascular disorders. Moreover, the conclusions in this study may assist in future investigations of the mechanisms underlying pulmonary endothelial dysfunction.

Both ECs and SMCs control vascular contractility and arterial pressure. The expression of Panx1 and TRPV4 channels in both ECs and SMCs (**Sharma et al., 2018**; **DeLalio et al., 2018**; **Martin et al., 2012**; **Ottolini et al., 2020a**; **Yang et al., 2006**) makes it challenging to decipher the cell type-specific roles of Panx1 and TRPV4 channels using global knockouts or pharmacological strategies. Indeed, global *Trpv4* knockout mice showed no systemic blood pressure or PAP phenotype (**Xia et al., 2013**; **Zhang et al., 2009**; **Hong et al., 2018**). However, inducible, *Trpv4* cKO-EC mice had elevated systemic blood pressure and PAP (**Daneva et al., 2021**; **Ottolini et al., 2020b**). Lack of a phenotype in global knockout mice could be due to the deletion of TRPV4 channels from multiple cell types or compensatory mechanisms that have developed over time (reviewed by **El-Brolosy and Stainier, 2017**). Therefore, studies utilizing cell-specific knockout mice are necessary for a definitive assessment of the control of PAP by EC and SMC Panx1 and TRPV4 channels. Although SMC TRPV4 channels have been shown to contribute to hypoxia-induced pulmonary vasoconstriction, resting PAP is not altered in global *Trpv4* knockout mice (**Xia et al., 2013**; **Yang et al., 2012**). Further, our studies indicate that SMC Panx1 and TRPV4 channels do not influence resting PAP. Taken together with findings from EC-knockout mice, these results provide strong evidence that endothelial, but not SMC, Panx1 and TRPV4 channels maintain low PA contractility and PAP under resting conditions. Despite the elevated PAP in EC-specific *Panx1*, *P2ry2*, and *Trpv4* cKO mice (**Daneva et al., 2021**), right ventricular hypertrophy was not observed. These findings could be attributed to a short duration of inducible genetic deletion in our studies. Although the duration of the knockout is sufficient to result in elevated PAP, a longer duration or larger changes in PAP may be required for observing right ventricular hypertrophy in these mouse models.

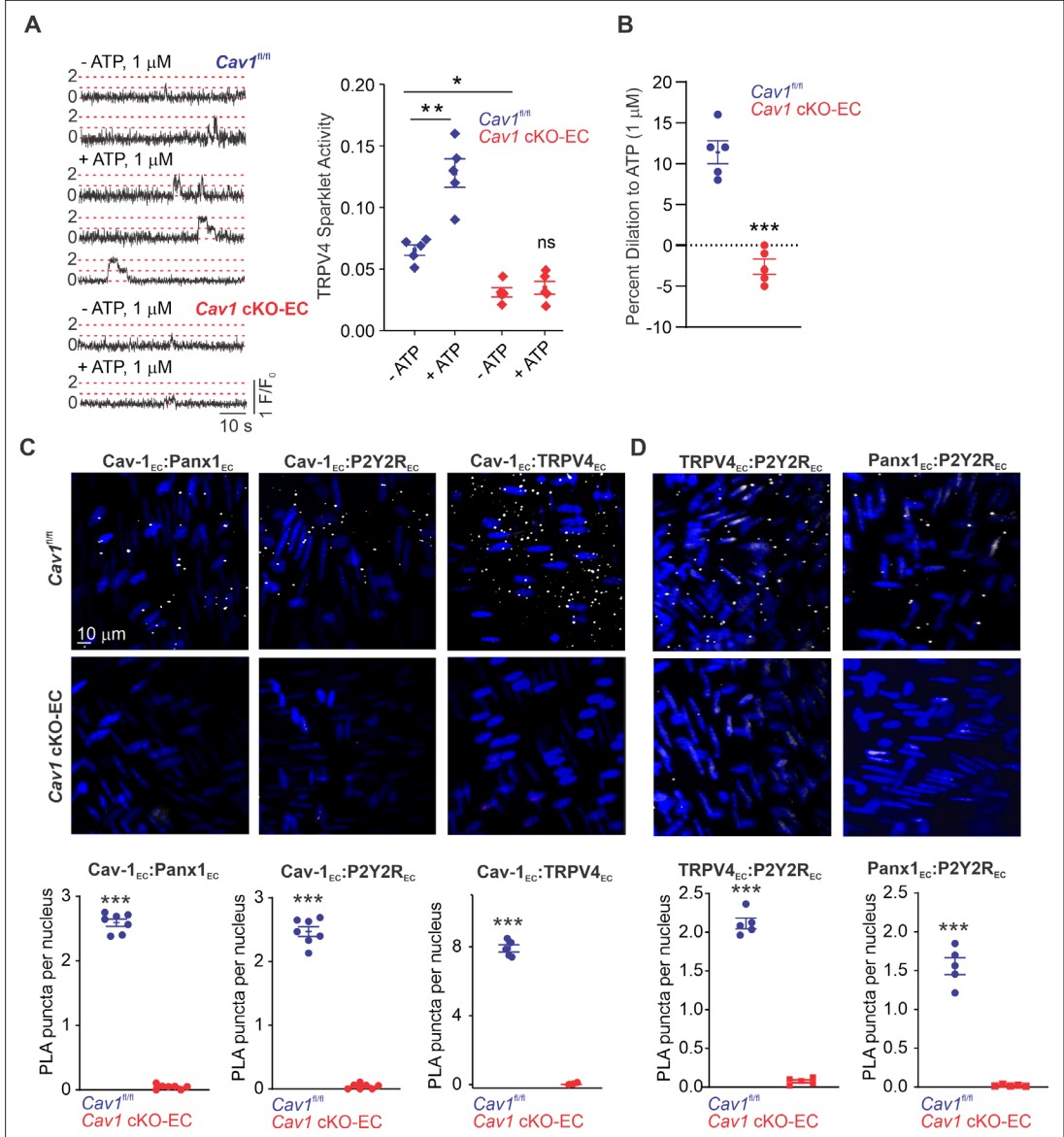

**Figure 4.** Cav-1$_{EC}$ provides a signaling scaffold for Panx1$_{EC}$–P2Y2R$_{EC}$–TRPV4$_{EC}$ signaling in pulmonary arteries (PAs). (**A**) Left: representative traces showing TRPV4$_{EC}$ sparklets in *en face* preparations of PAs from *Cav1*$^{fl/fl}$ and *Cav1* cKO-EC mice in the absence or presence of ATP (1 μmol/L). Dotted lines are quantal levels. Right: TRPV4$_{EC}$ sparklet activity (NP$_O$) per site in *en face* preparations of PAs from *Cav1*$^{fl/fl}$ and *Cav1* cKO-EC mice in the absence or presence of 1 μmol/L ATP (n = 5; *p<0.05 vs. *Cav1*$^{fl/fl}$ [- ATP]; **p<0.01 vs. *Cav1*$^{fl/fl}$ [- ATP]; ns indicates no statistical significance; two-way ANOVA). Experiments were performed in Fluo-4-loaded fourth-order PAs in the presence of cyclopiazonic acid (CPA; 20 μmol/L), included to eliminate Ca$^{2+}$ release from intracellular stores. 'N' is the number of channels per site and 'P$_O$' is the open state probability of the channel. (**B**) Percentage dilation of PAs from *Cav1*$^{fl/fl}$ and *Cav1* cKO-EC mice in response to ATP (1 μmol/L). PAs were pre-constricted with the thromboxane A2 receptor analog U46619 (50 nmol/L; n = 5; ***p<0.01 vs. *Cav1*$^{fl/fl}$; t-test). (**C**) Top: representative merged images of proximity ligation assays (PLAs) signal, showing EC nuclei and Cav-1$_{EC}$:Panx1$_{EC}$, Cav-1$_{EC}$:P2Y2R$_{EC}$, and Cav-1$_{EC}$:TRPV4$_{EC}$ co-localization (white puncta) in fourth-order PAs from *Cav1*$^{fl/fl}$ and *Cav1* cKO-EC mice. Bottom: quantification of Cav-1$_{EC}$:Panx1$_{EC}$, Cav-1$_{EC}$:P2Y2R$_{EC}$, and Cav-1$_{EC}$:TRPV4$_{EC}$ co-localization in PAs from *Cav1*$^{fl/fl}$ and *Cav1* cKO-EC mice (n = 5; ***p<0.001 vs. *Cav1*$^{fl/fl}$; t-test). (**D**) Representative PLA images showing EC nuclei, TRPV4$_{EC}$:P2Y2R$_{EC}$ and Panx1$_{EC}$:P2Y2R$_{EC}$ co-localization (white puncta) in fourth-order PAs from *Cav1*$^{fl/fl}$ and *Cav1* cKO-EC mice. Bottom: quantification of TRPV4$_{EC}$:P2Y2R$_{EC}$ and Panx1$_{EC}$:P2Y2R$_{EC}$ co-localization in PAs from *Cav1*$^{fl/fl}$ and *Cav1* cKO-EC mice (n = 5; ***p<0.001 vs. *Cav1*$^{fl/fl}$; t-test).

The online version of this article includes the following figure supplement(s) for figure 4:

**Figure supplement 1.** Representative proximity ligation assay (PLA) images showing EC nuclei, TRPV4$_{EC}$:P2Y2R$_{EC}$ and Panx1$_{EC}$:P2Y2R$_{EC}$ co-localization in fourth-order pulmonary arteries (PAs) from *P2ry2* cKO-EC mice.

**Figure supplement 2.** Left: representative proximity ligation assay (PLA) images showing EC nuclei and Cav-1$_{EC}$:P2Y1$_{EC}$ co-localization in fourth-order pulmonary arteries (PAs) from *Cav1*$^{fl/fl}$ mice.

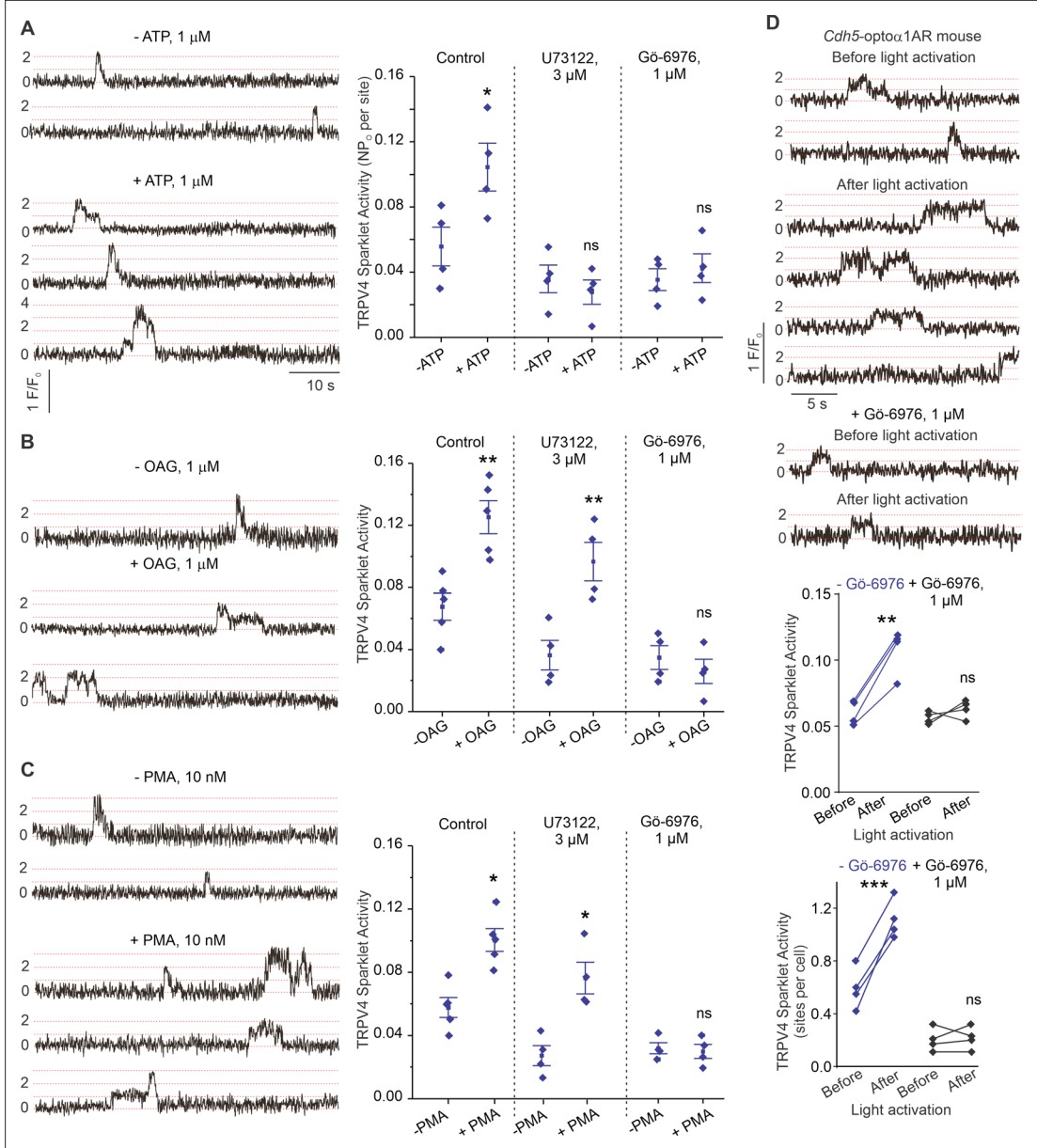

**Figure 5.** ATP activates TRPV4$_{EC}$ channels via phospholipase C–diacylglycerol–protein kinase C (PLC–DAG–PKC) signaling in pulmonary arteries (PAs). (**A**) Left: representative traces showing TRPV4$_{EC}$ sparklet activity in *en face* preparations of PAs from C57BL6/J mice before and after treatment with ATP (1 μmol/L). Right: effects of U73122 (PLC inhibitor; 3 μmol/L) or Gö-6976 (PKCα/β inhibitor; 1 μmol/L) on TRPV4$_{EC}$ sparklet activity in *en face* preparations of PAs from C57BL6/J mice before and after treatment with ATP (1 μmol/L), expressed as NP$_O$ per site. Experiments were performed in Fluo-4-loaded fourth-order PAs in the presence of cyclopiazonic acid (CPA; 20 μmol/L), included to eliminate Ca$^{2+}$ release from intracellular stores (n = 5; *p<0.05 vs. Control [-ATP]; ns indicates no statistical significance; one-way ANOVA). 'N' is the number of channels per site and 'P$_O$' is the open state probability of the channel. Dotted lines indicate quantal levels. (**B**) Left: representative traces showing TRPV4$_{EC}$ sparklet activity in *en face* preparations of PAs from C57BL6/J mice in the absence or presence of OAG (DAG analog; 1 μmol/L). Right: effects of U73122 (3 μmol/L) or Gö-6976 (1 μmol/L) on TRPV4$_{EC}$ sparklet activity in *en face* preparations of PAs from C57BL6/J mice before and after treatment with OAG (1 μmol/L, n = 6; **p<0.01 vs. Control [-OAG]; **p<0.01 vs. U73122 [-OAG]; ns indicates no statistical significance; one-way ANOVA). (**C**) Left: representative traces showing TRPV4$_{EC}$ sparklets in *en face* preparations of PAs from C57BL6/J mice in the absence or presence of phorbol myristate acetate (PMA) (PKC activator; 10 nmol/L). Right: effects of U73122 (3 μmol/L) or Gö-6976 (1 μmol/L) on TRPV4$_{EC}$ sparklet activity in *en face* preparations of PAs from C57BL6/J mice before and after treatment with PMA (n = 6; *p<0.05 vs. Control [-PMA]; *p<0.05 vs. U73122 [-PMA]; ns indicates no statistical significance; one-way ANOVA). (**D**) Top: representative traces showing TRPV4$_{EC}$ sparklet activity in *en face* preparations of PAs from *Cdh5*-optoα1AR (adrenergic receptor) mouse before and after light activation (470 nm). Center: scatterplot showing TRPV4 sparklet activity before and after light activation in the absence or presence of PKCα/β inhibitor Gö-6976 (1 μmol/L, n = 4, ***p<0.01 vs. –Gö-6976 [before]; ns indicates no statistical significance; one-way ANOVA). Bottom: scatterplot showing TRPV4 sparklet activity, expressed as sparklet sites per cell, before and after light activation, in the absence or presence of PKCα/β

*Figure 5 continued on next page*

Figure 5 continued

inhibitor Gö-6976 (1 µmol/L; n = 4; ***p<0.001 vs. –Gö-6976 [before]; ns indicates no statistical significance; one-way ANOVA).

The online version of this article includes the following figure supplement(s) for figure 5:

**Figure supplement 1.** A multi-Gaussian to all-points histogram obtained using sparklet traces from X-Rhod-1-loaded pulmonary arteries (PAs), showing quantal (evenly spaced) $\Delta F/F_0$ levels of 0.21.

**Figure supplement 2.** Left: scatterplot showing TRPV4 sparklet activity, expressed as $NP_O$ per site, before and after light activation, in the presence of TRPV4 inhibitor GSK2193874 (GSK219; 100 nmol/L, n = 4).

Recent studies in pulmonary fibroblasts and other cell types suggest that TRPV4 channel-mediated increases in cytosolic $Ca^{2+}$ can induce eATP release through Panx1 (**Baxter et al., 2014**; **Rahman et al., 2018**). However, the reverse interaction, in which Panx1-mediated eATP release activates TRPV4 channels, has not been explored in any cell type. Since Panx1 is activated by cytosolic $Ca^{2+}$ (**Locovei et al., 2006**) and eATP has been previously shown to activate $TRPV4_{EC}$ channels (**Marziano et al., 2017**), bidirectional signaling between Panx1 and TRPV4 channels is conceivable. Our demonstration that baseline eATP levels are unchanged in PAs from *Trpv4* cKO-EC mice rules out a role for $TRPV4_{EC}$ channels in controlling eATP release under baseline conditions. Moreover, $TRPV4_{EC}$ channels did not contribute to flow-induced efflux of ATP through $Panx1_{EC}$. Nevertheless, these data from pulmonary ECs do not rule out potential $TRPV4$–$Ca^{2+}$–Panx1 signaling in other cell types.

Elevated capillary $TRPV4_{EC}$ channel activity has been linked to increased endothelial permeability (**Thorneloe et al., 2012**; **Yin et al., 2008**), lung injury (**Alvarez et al., 2006**), and pulmonary edema (**Thorneloe et al., 2012**; **Yin et al., 2008**). Moreover, $Panx1_{EC}$-mediated eATP release is associated with vascular inflammation at the level of capillaries (**Sharma et al., 2018**). The physiological roles of $Panx1_{EC}$ and $TRPV4_{EC}$ channels in PAs, however, remain unknown. ECs from pulmonary capillaries and arteries are structurally and functionally different. Whereas PAs control pulmonary vascular resistance and PAP, capillaries control vascular permeability. $TRPV4_{EC}$ channels couple with distinct targets in arterial and capillary ECs (**Sonkusare et al., 2012**; **Longden et al., 2017**). Our data identify physiological roles of $Panx1_{EC}$–$TRPV4_{EC}$ channel signaling in PAs, but whether such signaling operates in the capillary endothelium and is essential for its physiological function is unclear.

Purinergic signaling and the endogenous purinergic receptor agonist eATP are essential controllers of pulmonary vascular function (**Konduri and Mital, 2000**; **Konduri et al., 2004**; **Hennigs et al., 2019**; **Kylhammar et al., 2014**). Our discovery of the $Panx1_{EC}$–$P2Y2R_{EC}$–$TRPV4_{EC}$ channel pathway establishes a signaling axis in ECs that regulates pulmonary vascular function. The pulmonary vasculature is a high-flow circulation, yet the flow-induced signaling mechanisms are poorly understood in PAs. Our results confirm that flow/shear stress increases ATP efflux through $Panx1_{EC}$ in PAs, which could be a potential mechanism for flow-induced dilation of PAs. Further investigations are needed to verify flow/shear stress-induced, eATP-dependent activation of $P2Y2R_{EC}$–$PKC\alpha$–$TRPV4_{EC}$ signaling in PAs. Several purinergic receptor subtypes are expressed in the pulmonary vasculature, including P2YRs and P2XRs (**Konduri et al., 2004**; **Hennigs et al., 2019**; **Syed et al., 2010**). Although only $P2Y2R_{EC}$ appears to mediate eATP activation of $TRPV4_{EC}$ channels, our studies do not rule out potentially important roles for other P2Y or P2X receptors in the pulmonary endothelium.

Activation of $TRPV4_{EC}$ channels by eATP released through $Panx1_{EC}$ in PAs would be facilitated by spatial localization of $TRPV4_{EC}$ channels with $Panx1_{EC}$. In keeping with this, several scaffolding proteins are known to promote localization of TRPV4 channels with their regulatory proteins, including A-kinase anchoring protein 150 (AKAP150) and Cav-1 (**Ottolini et al., 2020b**; **Li et al., 2018**). Although AKAP150 is not found in the pulmonary endothelium (**Marziano et al., 2017**), Cav-1 is a key structural protein in the pulmonary vasculature and has a well-established role in controlling $TRPV4_{EC}$ channel activity, pulmonary vascular function, and PAP (**Daneva et al., 2021**; **Zhao et al., 2002**; **Zhao et al., 2009**). Moreover, Cav-1-dependent signaling is impaired in pulmonary hypertension (**Daneva et al., 2021**; **Bakhshi et al., 2013**; **Maniatis et al., 2008**; **Nickel et al., 2015**). Studies in other cell types have shown that Cav-1 can co-localize with Panx1 and P2Y2Rs (**DeLalio et al., 2018**; **Martinez et al., 2016**). Additionally, Cav-1 can interact with PKC at the Cav-1 scaffolding domain (**Mineo et al., 1998**). Our results demonstrate that $Cav-1_{EC}$ exists in nanometer proximity with $Panx1_{EC}$, $P2Y2R_{EC}$, PKC, and $TRPV4_{EC}$ channels in PAs. Furthermore, the activation of $TRPV4_{EC}$ channels by $Panx1_{EC}$, eATP, $P2Y2R_{EC}$, or $PKC\alpha$ requires $Cav-1_{EC}$. Based on these findings, we conclude that $Cav-1_{EC}$ enables

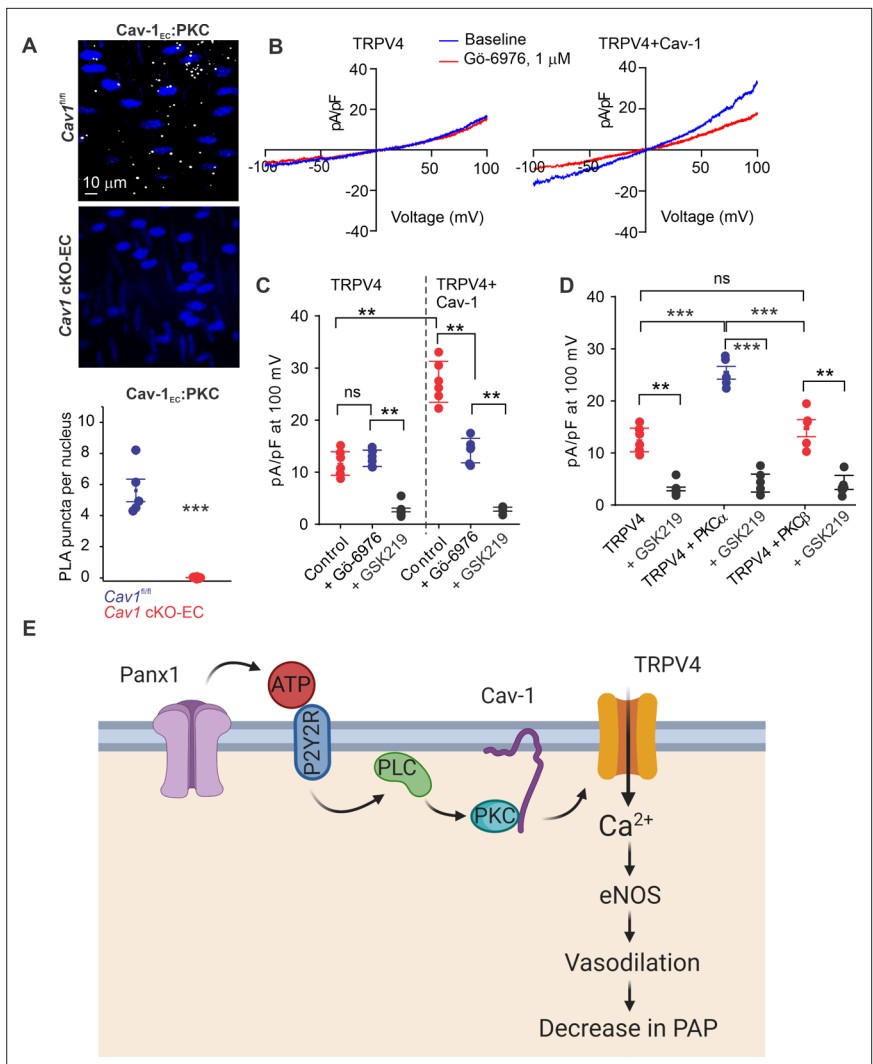

**Figure 6.** Localization of PKCα with Cav-1$_{EC}$ increases the activity of TRPV4$_{EC}$ channels in pulmonary arteries (PAs). (**A**) Top: representative merged images of proximity ligation assays (PLAs) showing endothelial cell (EC) nuclei and Cav-1$_{EC}$:PKC co-localization (white puncta) in fourth-order PAs from *Cav1*$^{fl/fl}$ and *Cav1* cKO-EC mice. Bottom: quantification of Cav-1$_{EC}$:PKC co-localization in PAs from *Cav1*$^{fl/fl}$ and *Cav1* cKO-EC mice (n = 5; ***p<0.001 vs. *Cav1*$^{fl/fl}$; t-test). (**B**) Representative traces showing TRPV4 currents in the absence or presence of Gö-6976 (PKC inhibitor; 1 μmol/L) in HEK293 cells transfected with TRPV4 alone or co-transfected with TRPV4 plus wild-type Cav-1, recorded in the whole-cell patch-clamp configuration. (**C**) Current density scatterplot of TRPV4 currents at +100 mV in the absence or presence of Gö-6976 (1 μmol/L) and after the addition of GSK2193874 (GSK219; TRPV4 inhibitor; 100 nmol/L) in HEK293 cells transfected with TRPV4 alone or TRPV4 plus wild-type Cav-1 (n = 5; **p<0.01 vs. Control [TRPV4]; **p<0.01 vs. Control [TRPV4+ Cav-1]; ns indicates no statistical significance; one-way ANOVA). (**D**) Current density plot of TRPV4 currents at +100 mV in HEK293 cells transfected with TRPV4+ PKCα or TRPV4+ PKC $\beta$ and in the presence of GSK219 (100 nmol/L; n = 5; ***p<0.001 vs. TRPV4+ PKCα; t-test). (**E**) Schematic depiction of the Panx1$_{EC}$–P2Y2R$_{EC}$–TRPV4$_{EC}$ signaling pathway that promotes vasodilation and lowers pulmonary arterial pressure (PAP) in PAs. ATP released from Panx1$_{EC}$ activates P2Y2R$_{EC}$ purinergic receptors on the EC membrane. Stimulation of P2Y2R$_{EC}$ recruits PKCα, which anchors to the scaffolding protein Cav-1$_{EC}$ in close proximity to TRPV4$_{EC}$ channels. TRPV4$_{EC}$ channel-dependent vasodilation lowers PAP.

Panx1$_{EC}$–P2Y2R$_{EC}$–TRPV4$_{EC}$ signaling at EC membranes in PAs. Cav-1 is also a well-known anchor protein for eNOS (*Bernatchez et al., 2005*), acting by stabilizing eNOS expression and negatively regulating its activity (*Bernatchez et al., 2005*). We previously showed that TRPV4$_{EC}$ Ca$^{2+}$ sparklets activate eNOS in PAs (*Marziano et al., 2017*; *Ottolini et al., 2020a*). Specifically, TRPV4 channel activation increased endothelial NO levels, an effect that was absent in PAs from *eNOS* knockout

mice (**Marziano et al., 2017**). Moreover, TRPV4 channel-induced vasodilation was abolished by NOS inhibitor L-NNA. Thus, Cav-1$_{EC}$ enhancement of Ca$^{2+}$ influx through TRPV4$_{EC}$ channels may represent novel mechanisms for regulating eNOS activity.

Cav-1$_{EC}$/PKCα-dependent signaling is a novel endogenous mechanism for activating arterial TRPV4$_{EC}$ channels and lowering PAP. Proximity to PKCα appears to be crucial for the normal function of TRPV4 channels. Evidence from the systemic circulation suggests that co-localization of TRPV4 channels with scaffolding proteins enhances their activity (**Mercado et al., 2014**; **Sonkusare et al., 2014**), and we specifically demonstrated that PKC anchoring by AKAP150 enhances the activity of TRPV4$_{EC}$ channels in mesenteric arteries (**Ottolini et al., 2020b**). Here, we show that PKC anchoring by Cav-1$_{EC}$ enables PKC activation of TRPV4$_{EC}$ channels in PAs. This discovery raises the possibility that disruption of PKC anchoring by Cav-1$_{EC}$ could impair the Panx1$_{EC}$–P2Y2R$_{EC}$–TRPV4$_{EC}$ signaling axis under disease conditions. A lack of PKC anchoring by scaffolding proteins in systemic arteries has been demonstrated in obesity and hypertension (**Ottolini et al., 2020b**; **Sonkusare et al., 2014**). Further studies of pulmonary vascular disorders are required to establish whether the Panx1$_{EC}$–P2Y2R$_{EC}$–PKCα–TRPV4$_{EC}$ signaling axis is impaired in pulmonary vascular disorders.

In conclusion, Panx1$_{EC}$–P2Y2R$_{EC}$–TRPV4$_{EC}$ channel signaling reduces PA contractility and maintains a low resting PAP. This mechanism is facilitated by eATP released through Panx1$_{EC}$ and subsequent activation of P2Y2R$_{EC}$–PKCα signaling. Cav-1$_{EC}$ ensures the spatial proximity among Panx1$_{EC}$, P2Y2R$_{EC}$, and TRPV4$_{EC}$ channels and also anchors PKCαclose to TRPV4$_{EC}$ channels. These findings identify a novel endothelial Ca$^{2+}$ signaling mechanism that reduces PA contractility. Further investigations are needed to determine whether impairment of this pathway contributes to elevated PAP in pulmonary vascular disorders and whether this pathway can be targeted for therapeutic benefit.

# Materials and methods

## Key resources table

| Reagent type (species) or resource | Designation | Source or reference | Identifiers | Additional information |
|---|---|---|---|---|
| Genetic reagent (*Mus musculus*) | C57BL/6J | The Jackson Laboratory | Stock no: 000664 | |
| Genetic reagent (*M. musculus*) | *Trpv4* conditional knockout in EC | Dr. Swapnil Sonkusare PMID:32008372 | | |
| Genetic reagent (*M. musculus*) | *Trpv4* conditional knockout in SMC | Dr. Swapnil Sonkusare PMID:33879616 | | |
| Genetic reagent (*M. musculus*) | *Panx1* conditional knockout in EC | Dr. Brant Isakson PMID:26242575 | | |
| Genetic reagent (*M. musculus*) | *Panx1* conditional knockout in SMC | Dr. Brant Isakson PMID:25690012 | | |
| Genetic reagent (*M. musculus*) | *Cav1* conditional knockout in EC | Dr. Swapnil Sonkusare PMID:33879616 Dr. Richard Minshall PMID:22323292 | | |
| Genetic reagent (*M. musculus*) | *P2ry2*$^{fl/fl}$ mice | Dr. Cheikh Seye PMID:27856454 | | |
| Genetic reagent (*M. musculus*) | Cdh5-Opto α 1AR-IRES-lacZ | CHROMus (Cornell University, USA) | | |
| Antibody | TRPV4 antibody (aa100-150), (mouse polyclonal) | LifeSpan Bioscience Inc | Cat. #: LS-C94498; RRID:AB_2893149 | (1:200) |
| Antibody | Anti-caveolin-1 antibody - caveolae marker (rabbit polyclonal) | Abcam plc | Cat. #: Ab2910; RRID:AB_303405 | (1:500) |
| Antibody | Caveolin-1 antibody (7C8) (mouse monoclonal) | Novus Biologicals, LLC | Cat. #: NB100-615; RRID:AB_10003431 | (1:200) |
| Antibody | PKC (mouse monoclonal) | Santa Cruz Biotechnology, Inc | Cat. #: SC-17769; RRID:AB_628139 | (1:250) |

*Continued on next page*

*Continued*

| Reagent type (species) or resource | Designation | Source or reference | Identifiers | Additional information |
|---|---|---|---|---|
| Antibody | Panx1 (rabbit polyclonal) | Alomone Labs | Cat. #: ACC-234; RRID:AB_2340917 | (1:100) |
| Antibody | P2Y2R (rabbit polyclonal) | Alomone Labs | Cat. #: APR-010; RRID:AB_2040078 | (1:250) |
| Antibody | P2Y1R (rabbit polyclonal) | Alomone Labs | Cat. #: APR-009; RRID:AB_2040070 | (1:100) |
| Chemical compound, drug | GSK2193874 | Tocris Bioscience | Cat. #: 5106/5 | |
| Chemical compound, drug | Cyclopiazonic acid (CPA) | Tocris Bioscience | Cat. #: 1235/10 | |
| Chemical compound, drug | GSK1016790A | Tocris Bioscience | Cat. #: 6433/10 | |
| Chemical compound, drug | Phorbol 12-myristate 13-acetate (PMA) | Tocris Bioscience | Cat. #: 1201/1 | |
| Chemical compound, drug | AR-C 118925XX | Tocris Bioscience | Cat. #: 4890/5 | |
| Chemical compound, drug | 2-Thio UTP tetrasodium salt | Tocris Bioscience | Cat. #: 3280/1 | |
| Chemical compound, drug | MRS2179 | Tocris Bioscience | Cat. #: 0900/10 | |
| Chemical compound, drug | U-73122 | Tocris Bioscience | Cat. #: 1268/10 | |
| Chemical compound, drug | NS309 | Tocris Bioscience | Cat. #: 3895/10 | |
| Chemical compound, drug | ARL-67156 | Tocris Bioscience | Cat. #: 1283/10 | |
| Other | Fluo-4-AM | Invitrogen | Cat. #: F14201 | |
| Chemical compound, drug | 1-O-9Z-octadecenoyl-2-O-acetyl-*sn*-glycerol (OAG) | Cayman Chemicals | Cat. #: 62600 | |
| Chemical compound, drug | PPADS | Cayman Chemicals | Cat. #: 14537 | |
| Chemical compound, drug | Gö-6976 | Cayman Chemicals | Cat. #: 13310 | |
| Chemical compound, drug | JNJ-47965567 | Cayman Chemicals | Cat. #: 21895 | |
| Chemical compound, drug | U46619 | Cayman Chemicals | Cat. #: 16452 | |
| Chemical compound, drug | Tamoxifen | Sigma-Aldrich | Cat. #: T5648 | |
| Peptide, recombinant protein | Apyrase | Sigma-Aldrich | Cat. #: A6535 | |
| Software, algorithm | LabChart8 | ADInstruments https://www.adinstruments.com/products/labchart | RRID:SCR_017551 | |
| Software, algorithm | Segment version 2.0 R5292 | Twilio (http://segment.heiberg.se) | | |
| Software, algorithm | IonOptix | IonOptix, LLC ( https://www.ionoptix.com/products/software/ionwizard-core-and-analysis/) | | |
| Software, algorithm | SparkAn | Dr. Adrian Bonev, University of Vermont, Burlington, VT, USA PMID:22556255 | | |
| Software, algorithm | ClampFit10.3 | Molecular Devices (https://www.moleculardevices.com/) | RRID:SCR_011323 | |
| Software, algorithm | ImageJ | National Institutes of Health (https://imagej.nih.gov/ij/) | RRID:SCR_003070 | |

*Continued on next page*

*Continued*

| Reagent type (species) or resource | Designation | Source or reference | Identifiers | Additional information |
|---|---|---|---|---|
| Software, algorithm | PatchMaster v2x90 program | Harvard Bioscience https://www.harvard bioscience.com/ | RRID:SCR_000034 | |
| Software, algorithm | FitMaster v2x73.2 | Harvard Bioscience https://www.harvard bioscience.com/ | RRID:SCR_016233 | |
| Software, algorithm | MATLAB R2018a | MathWorks https://www.mathworks. com/products/matlab.html | RRID:SCR_013499 | |
| Software, algorithm | CorelDraw Graphics Suite X7 | CorelDraw (https://www.coreldraw.com/en) | RRID:SCR_014235 | |
| Software, algorithm | GraphPad Prism 8.3.0 | GraphPad Software, Inc (https://www.graphpad.com/) | RRID:SCR_002798 | |
| Software, algorithm | GLIMMPSE software | (https://glimmpse.samplesizeshop. org/) | RRID:SCR_016297 | |
| Software, algorithm | Biorender | http://biorender.com | RRID:SCR_018361 | |

## Drugs and chemical compounds

Cyclopiazonic acid (CPA), GSK2193874, GSK1016790A, phorbol 12-myristate 13-acetate (PMA), AR-C 118925XX , 2-Thio UTP tetrasodium salt, MRS2179, U-73122, NS309, and ARL-67156 were purchased from Tocris Bioscience (Minneapolis, MN). Fluo-4-AM ($Ca^{2+}$ indicator) were purchased from Invitrogen (Carlsbad, CA). 1-O-9Z-octadecenoyl-2-O-acetyl-*sn*-glycerol (OAG), PPADS (sodium salt), Gö-6976, JNJ-47965567, and U46619 were purchased from Cayman Chemicals (Ann Arbor, MI). Tamoxifen and apyrase were obtained from Sigma-Aldrich (St. Louis, MO).

## Animal protocols and models

All animal protocols were approved by the University of Virginia Animal Care and Use Committee (protocols 4100 and 4120). Both male and female mice were used in this study and age- and sex-matched controls were used. No sex differences were observed in RVSPs and TRPV4-induced dilation of PAs. C57BL6/J were obtained from the Jackson Laboratory (Bar Harbor, ME). Inducible endothelial cell (EC)-specific TRPV4 channel knockout (*Trpv4* cKO-EC; *Lohman et al., 2015*; *Moore et al., 2013*), smooth muscle cell (SMC)-specific TRPV4 channel knockout (*Trpv4* cKO-SMC; *Billaud et al., 2015*), EC-specific caveolin-1 knockout (*Cav1* cKO-EC; *Chen et al., 2012*), EC-specific P2Y2R receptor knockout (*P2ry2* cKO-EC; *Chen et al., 2017*), EC-specific Panx1 channel knockout (*Panx1* cKO-EC; *Lohman et al., 2015*; *Poon et al., 2014*) and SMC-specific Panx1 channel knockout (*Panx1* cKO-SMC; *Billaud et al., 2015*) mice (10–14 weeks old) were used. Mice were housed in an enriched environment and maintained under a 12:12 hr light/dark photocycle at ~23 °C with fresh tap water and standard chow diet available ad libitum. Mice were euthanized with pentobarbital (90 mg/kg; intraperitoneally; Diamondback Drugs, Scottsdale, AZ) followed by cervical dislocation for harvesting lung tissue. Fourth-order PAs (~50 µm diameter) were isolated in cold HEPES-buffered physiological salt solution (HEPES-PSS, in mmol/L, 10 HEPES, 134 NaCl, 6 KCl, 1 $MgCl_2$ hexahydrate, 2 $CaCl_2$ dihydrate, and 7 dextrose, pH adjusted to 7.4 using 1 mol/L NaOH).

$Trpv4^{fl/fl}$ (*Moore et al., 2013*), $Cav1^{fl/fl}$ (*Chen et al., 2012*), $Panx1^{fl/fl}$ (*Lohman et al., 2015*; *Poon et al., 2014*) and $P2ry2^{fl/fl}$ [57] mice were crossed with VE-cadherin (*Cdh5*, endothelial) Cre mice (*Moore et al., 2013*) or *SMMHC* (smooth muscle) Cre mice (*Wirth et al., 2008*). EC- or SMC-specific knockout of *Trpv4*, *Cav1*, *Panx1*, or *P2ry2* was induced by injecting 6-week-old $Trpv4^{fl/fl}$ Cre$^+$, $Cav1^{fl/fl}$ Cre$^+$, $Panx1^{fl/fl}$ Cre$^+$, and $P2ry2^{fl/fl}$ Cre$^+$ mice with tamoxifen (40 mg/kg intraperitoneally per day for 10 days). Tamoxifen-injected $Trpv4^{fl/fl}$ Cre$^-$, $Cav1^{fl/fl}$ Cre$^-$, $Panx1^{fl/fl}$ Cre$^-$, and $P2ry2^{fl/fl}$ Cre$^-$ mice were used as controls. Mice were used for experiments after a 2-week washout period. Genotypes for *Cdh5* Cre and *SMMHC* Cre were confirmed following previously published protocols (*Moore et al., 2013*; *Wirth et al., 2008*). $Trpv4^{fl/fl}$ (*Moore et al., 2013*), $Cav1^{fl/fl}$ (*Chen et al., 2012*), $Panx1^{fl/fl}$ (*Lohman et al., 2015*;

*Poon et al., 2014*), and *P2ry2*<sup>fl/fl</sup> (*Chen et al., 2017*) genotyping was performed as previously. *Cdh5*-Optoα1AR mice were developed by CHROMus (Cornell University, USA).

## RVSP and Fulton index measurement

Mice were anesthetized with pentobarbital (50 mg/kg bodyweight; intraperitoneally) and bupivacaine HCl (100 µL of 0.25% solution; subcutaneously) was used to numb the dissection site on the mouse. RVSP was measured as an indirect indicator of PAP. A Mikro-Tip pressure catheter (SPR-671; Millar Instruments, Huston, TX), connected to a bridge amp (FE221), and a PowerLab 4/35 4-channel recorder (ADInstruments, Colorado Springs, CO), was inserted through the external jugular vein into the right ventricle. Right ventricular pressure and heart rate were acquired and analyzed using LabChart8 software (ADInstruments). A stable 3 min recording was acquired for all the animals, and 1 min continuous segment was used for data analysis. When necessary, traces were digitally filtered using a low-pass filter at a cutoff frequency of 50 Hz. At the end of the experiments, mice were euthanized, and the hearts were isolated for right ventricular hypertrophy analysis. Right ventricular hypertrophy was determined by calculating the Fulton index, a ratio of the right ventricular (RV) heart weight over the left ventricular (LV) plus septum (S) weight (RV/ LV + S).

## Luciferase assay for total ATP release

ATP assay protocol was adapted from *Yang et al., 2020*. Fourth-order PAs (~50 µm diameter) were isolated in cold HEPES-buffered physiological salt solution (HEPES-PSS, in mmol/L, 10 HEPES, 134 NaCl, 6 KCl, 1 MgCl$_2$ hexahydrate, 2 CaCl$_2$ dihydrate, and 7 dextrose, pH adjusted to 7.4 using 1 mol/L NaOH). Isolated PAs were pinned down *en face* on a Sylgard block and cut open. PAs were placed in black, opaque 96-well plates and incubated in HEPES-PSS for 10 min at 37 °C, followed by incubation with the ectonucleotidase inhibitor ARL 67156 (300 µmol/L, Tocris Bioscience, Minneapolis, MN) for 30 min at 37 °C. 50 µL volume of each sample was transferred to another black, opaque 96-well plate. ATP was measured using ATP bioluminescence assay reagent ATP Bioluminescence HSII kit (Roche Applied Science, Penzberg, Germany). Using a luminometer (FluoStar Omega), 50 µL of luciferin:luciferase reagent (ATP bioluminescence assay kit HSII; Roche Applied Science) was injected into each well and luminescence was recorded following a 5 s orbital mix and sample measurement at 7 s. ATP concentration in each sample was calculated from an ATP standard curve. For some experimental groups, PAs were first mounted on a pressure myography chamber and were denuded by pushing air through the lumen for 1 min.

## Cardiac magnetic resonance imaging (MRI)

MRI studies were conducted under protocols that comply with the Guide for the Care and Use of Laboratory Animals (NIH publication no. 85-23, revised 1996). Mice were positioned in the scanner under 1.25% isoflurane anesthesia and body temperature was maintained at 37 °C using thermostatic circulating water. A cylindrical birdcage RF coil (30 mm diameter, Bruker, Ettlingen, Germany) with an active length of 70 mm was used, and heart rate, respiration, and temperature were monitored during imaging using a fiber optic, MR-compatible system (Small Animal Imaging Inc, Stony Brook, NY). MRI was performed on a 7 Tesla (T) Clinscan system (Bruker) equipped with actively shielded gradients with a full strength of 650 mT/m and a slew rate of 6666 mT/m/ms (*Vandsburger et al., 2007*). Six short-axis slices were acquired from base to apex, with slice thickness of 1 mm, in-plane spatial resolution of 0.2 × 0.2 mm$^2$, and temporal resolution of 8–12 ms. Baseline ejection fraction (EF), end-diastolic volume (EDV), end-systolic volume (ESV), myocardial mass, wall thickness, stroke volume (SV), and cardiac output (CO) were assessed from the cine images using the freely available software Segment version 2.0 R5292 (http://segment.heiberg.se).

## Pressure myography

Isolated mouse PAs (~50 µm) were cannulated on glass micropipettes in a pressure myography chamber (The Instrumentation and Model Facility, University of Vermont, Burlington, VT) at areas lacking branching points and were pressurized at a physiological pressure of 15 mm Hg (*Ottolini et al., 2020a*). Arteries were superfused with PSS (in mmol/L, 119 NaCl, 4.7 KCl, 1.2 KH$_2$PO$_4$, 1.2 MgCl$_2$ hexahydrate, 2.5 CaCl$_2$ dihydrate, 7 dextrose, and 24 NaHCO$_3$) at 37 °C and bubbled with 20% O$_2$/5% CO$_2$ to maintain the pH at 7.4. All drug treatments were added to the superfusing PSS. PAs

were pre-constricted with 50 nmol/L U46619 (a thromboxane A2 receptor agonist). All other pharmacological treatments were performed in the presence of U46619. Before measurement of vascular reactivity, arteries were treated with NS309 (1 μmol/L), a direct opener of endothelial IK/SK channels, to assess endothelial health. Arteries that failed to fully dilate to NS309 were discarded. Changes in arterial diameter were recorded at a 60-ms frame rate using a charge-coupled device camera and edge-detection software (IonOptix LLC, Westwood, MA; *Sonkusare et al., 2012*; *Sonkusare et al., 2014*). All drug treatments were incubated for 10 min. At the end of each experiment, $Ca^{2+}$-free PSS (in mmol/L, 119 NaCl, 4.7 KCl, 1.2 $KH_2PO_4$, 1.2 $MgCl_2$ hexahydrate, 7 dextrose, 24 $NaHCO_3$, and 5 EGTA) was applied to assess the maximum passive diameter. Percent constriction was calculated by

$$[(\text{Diameter}_{\text{before}}\text{Diameter}_{\text{after}})/\text{Diameter}_{\text{before}}]100 \tag{1}$$

where $\text{Diameter}_{\text{before}}$ is the diameter of the artery before a treatment and $\text{Diameter}_{\text{after}}$ is the diameter after the treatment. Percent dilation was calculated by

$$[(\text{Diameter}_{\text{dilated}}\text{Diameter}_{\text{basal}})/(\text{Diameter}_{\text{Cafree}}\text{Diameter}_{\text{basal}})]100 \tag{2}$$

w here $\text{Diameter}_{\text{basal}}$ is the stable diameter before drug treatment, $\text{Diameter}_{\text{dilated}}$ is the diameter after drug treatment, and $\text{Diameter}_{\text{Ca-free}}$ is the maximum passive diameter.

## Flow/shear stress-induced ATP release

Flow/shear stress was measured using a protocol modified from *Ahn et al., 2017*. Briefly, isolated PAs (~50 μm) were cannulated on glass micropipettes in a pressure myography chamber (The Instrumentation and Model Facility, University of Vermont) at areas lacking branching points and were pressurized at a physiological pressure of 15 mm Hg (*Ottolini et al., 2020a*). Arteries were superfused with PSS (in mmol/L, 119 NaCl, 4.7 KCl, 1.2 $KH_2PO_4$, 1.2 $MgCl_2$ hexahydrate, 2.5 $CaCl_2$ dihydrate, 7 dextrose, and 24 $NaHCO_3$) at 37 °C and bubbled with 20% $O_2$/5% $CO_2$ to maintain the pH at 7.4. The arteries were treated luminally with 300 μmol/L ARL-67156 (ecto-ATPase inhibitor; Sigma-Aldrich) to avoid ATP degradation throughout the duration of the experiment. The tips of the cannulating pipettes were always arranged with smaller pipettes upstream and larger pipettes downstream. The average tip size was 20.1 ± 0.4 μm at the upstream end and 23.6 ± 0.4 μm at the downstream end. Both ends of the vessel were secured, and the vessel was maintained at an intraluminal pressure of 15 $cmH_2O$ by elevating the inflow reservoir. Flow/shear stress was increased by adjusting the height of the reservoir. Flow-induced luminal solution was collected at the outflow pipette end. After a 30 min equilibration period, a baseline sample was collected for luminal ATP measurement. Shear stress was calculated from the flow rate in the vessel lumen and the diameter of the vessels using the equation (*Zemskov et al., 2011*) : $\tau = 4(\mu \dot{Q}/(\pi r^3))$, where μ is viscosity, $\dot{Q}$ is volumetric flow rate, and $r$ is internal radius of the vessel. The volumetric flow rate was measured as the volume of the flowthrough at different pressures. Vessel diameter was measured at each flow rate. The shear stress range was 4–14 dynes/$cm^2$. Luminal outflow samples per shear stress range were obtained every 30 min. The samples were used for luciferase assays for total ATP release, as described above.

## $Ca^{2+}$ imaging

Measurements of $TRPV4_{EC}$ $Ca^{2+}$ sparklets in the native endothelium of mouse PAs were performed as previously described (*Sonkusare et al., 2012*). Briefly, fourth-order (~50 μm) PAs were pinned down *en face* on a Sylgard block and loaded with Fluo-4-AM (10 μmol/L) in the presence of pluronic acid (0.04%) at 30 °C for 30 min. $TRPV4_{EC}$ $Ca^{2+}$ sparklets were recorded at 30 frames per second with Andor Revolution WD (with Borealis) spinning-disk confocal imaging system (Oxford Instruments, Abingdon, UK) comprised an upright Nikon microscope with a 60× water dipping objective (numerical aperture 1.0) and an electron multiplying charge coupled device camera (iXon 888, Oxford Instruments). All experiments were carried out in the presence of cyclopiazonic acid (20 μmol/L, a sarco-endoplasmic reticulum [ER] $Ca^{2+}$-ATPase inhibitor) in order to eliminate the interference from $Ca^{2+}$ release from intracellular stores. Fluo-4 was excited at 488 nm with a solid-state laser and emitted fluorescence was captured using a 525/36 nm band-pass filter. $TRPV4_{EC}$ $Ca^{2+}$ sparklets were recorded before and 5 min after the addition of specific compounds. To generate fractional fluorescence ($F/F_0$) traces, a

region of interest defined by a 1.7-µm² (5 × 5 pixels) box was placed at a point corresponding to peak sparklet amplitude. Each field of view was ~110 × 110 µm and covered ~15 ECs. Representative F/F₀ traces were filtered using a Gaussian filter and a cutoff corner frequency of 4 Hz. Sparklet activity was assessed as described previously using the custom-designed SparkAn software (*Sonkusare et al., 2012*; *Sonkusare et al., 2014*).

## Calculation of TRPV4 sparklet activity per site

Activity of TRPV4 Ca²⁺ sparklets was analyzed as described previously (*Sonkusare et al., 2012*; *Ottolini et al., 2020b*; *Sonkusare et al., 2014*). Area under the curve for all the events at a site was determined using trapezoidal numerical integration ([F−F₀]/F₀ over time, in seconds). The average number of active TRPV4 channels, as defined by NP$_O$ (where N is the number of channels at a site and P$_O$ is the open state probability of the channel), was calculated by

$$NP_O = (T_{level1} + 2T_{level2} + 3T_{level3} + 4T_{level4})/T_{total} \qquad (3)$$

where T is the dwell time at each quantal level detected at TRPV4 sparklet sites and T$_{total}$ is the duration of the recording. NP$_O$ was determined using Single Channel Search module of Clampfit and quantal amplitudes derived from all-points histograms (*Marziano et al., 2017*) ($\Delta F/F_0$ of 0.29 for Fluo-4-loaded PAs).

Total number of sparklet sites in a field was divided by the number of cells in that field to obtain sparklet sites per cell.

## All-points histograms

All-points amplitude histograms were constructed as described previously (*Sonkusare et al., 2012*; *Ottolini et al., 2020b*). Briefly, images were filtered with a Kalman filter (adopted from an ImageJ plug-in written by Christopher Philip Mauer, Northwestern University, Chicago, IL; acquisition noise variance estimate = 0.05; filter gain = 0.8). The inclusion criteria were a stable baseline containing at least five steady points and a steady peak containing at least five peak points. Sparklet traces were exported to ClampFit10.3 for constructing an all-points histogram, which was fit with the multiple Gaussian function below:

$$f\left(F/F_0\right) = \sum_{i=1}^{N} \frac{a_i}{\sqrt{2\pi}\sigma_i} exp\left[\frac{-\left(\frac{F}{F_0}-\mu_i\right)^2}{2\sigma_i^2}\right] \qquad (4)$$

where F/F₀ represents fractional fluorescence, *a* represents the area, µ represents the mean value, and σ2 represents the variance of the Gaussian distribution. While the detected sparklets can have multiple amplitudes corresponding to quantal level 1, 2, 3, or 4, the baseline (level 0) was the same for all the detected sparklets regardless of the amplitude of the sparklets. Therefore, the baseline corresponds to a higher count compared to all other events.

## Immunostaining

Immunostaining was performed on fourth-order PAs (~50 µm) pinned *en face* on SYLGARD blocks. PAs were fixed with 4% paraformaldehyde (PFA) at room temperature for 15 min and then washed three times with phosphate-buffered saline (PBS). The tissue was permeabilized with 0.2% Triton-X for 30 min, blocked with 5% normal donkey serum (ab7475, Abcam, Cambridge, MA) or normal goat serum (ab7475, Abcam), depending on the host of the secondary antibody used, for 1 hr at room temperature. PAs were incubated with the primary antibodies (Key resources table) overnight at 4 °C. Following the overnight incubation, PAs were incubated with secondary antibody 1:500 Alexa Fluor 568-conjugated donkey anti-rabbit (Life Technologies, Carlsbad, CA) for 1 hr at room temperature in the dark room. For nuclear staining, PAs were washed with PBS and then incubated with 0.3 mmol/L DAPI (Invitrogen, Carlsbad, CA) for 10 min at room temperature. Images were acquired along the z-axis from the surface of the endothelium to the bottom where the EC layer encounters the smooth muscle cell layer with a slice size of 0.1 µm using the Andor microscope described above. The internal elastic lamina (IEL) autofluorescence was evaluated using an excitation of 488 nm with a solid-state

laser and collecting the emitted fluorescence with a 525/36 nm band-pass filter. Immunostaining for the protein of interest was evaluated using an excitation of 561 nm and collecting the emitted fluorescence with a 607/36 nm band-pass filter. DAPI immunostaining was evaluated using an excitation of 409 nm and collecting the emitted fluorescence with a 447/69 nm band-pass filter. The specificity of Panx1 and P2Y2R antibodies was confirmed by a lack of signal in PAs from endothelial knockout mice. The specificity of TRPV4, Cav-1, and PKC antibodies was confirmed previously (*Daneva et al., 2021*; *Ottolini et al., 2020b*).

## In situ PLA

Fourth-order (~50 µm) PAs were pinned *en face* on SYLGARD blocks. PAs were fixed with 4% PFA for 15 min followed by three washes with PBS. PAs were then permeabilized with 0.2% Triton X for 30 min at room temperature followed by blocking with 5% normal donkey serum (Abcam plc, Cambridge, MA) and 300 mmol/L glycine for 1 hr at room temperature. After three washes with PBS, PAs were incubated with the primary antibodies (Key resources table) overnight at 4 °C. The PLA protocol from Duolink PLA Technology kit (Sigma-Aldrich) was followed for the detection of co-localized proteins. Lastly, PAs were incubated with 0.3 µmol/L DAPI nuclear staining (Invitrogen) for 10 min at room temperature in the dark room. PLA images were acquired using the Andor Revolution spinning-disk confocal imaging system along the z-axis at a slice size of 0.1 µm. Images were analyzed by normalizing the number of positive puncta by the number of nuclei in a field of view. The specificity of the PLA antibodies was determined using PAs from endothelial knockout mice for one of the protein pairs.

## Plasmid generation and transfection into HEK293 cells

HEK293 cells authenticated with STR profiling were obtained from ATCC USA. Mycoplasma contamination was not detected as per ATCC website. The TRPV4 coding sequence without stop codons was amplified from mouse heart cDNA. The amplified fragment was inserted into a plasmid backbone containing a CMV promoter region for expression and, in addition, is suitable for lentiviral production by Gibson assembly. The in-frame FLAG tag was inserted into the 3′-primer used for amplification. Constructs were verified by sequencing the regions that had been inserted into the plasmid backbone. HEK293 cells were seeded ($7 \times 10^5$ cells per 100 mm dish) in Dulbecco's Modified Eagle Medium with 10% fetal bovine serum (Thermo Fisher Scientific Inc, Waltham, MA) 1 day prior to transfection. Cells were transfected using the LipofectamineLTX protocol (Thermo Fisher Scientific Inc). TRPV4 was co-expressed with PKCα and PKCβ, obtained from Origene Technologies (Montgomery County, MD).

## Patch clamp in freshly isolated pulmonary ECs and in HEK293 cells

Fresh ECs were obtained via enzymatic digestion of fourth-order PAs. Briefly, PAs were incubated in the dissociation solution (in mmol/L, 55 NaCl, 80 Na glutamate, 6 KCl, 2 $MgCl_2$, 0.1 $CaCl_2$, 10 glucose, 10 HEPES, pH 7.3) containing Worthington neutral protease (0.5 mg/mL) for 30 min at 37 °C. The extracellular solution consisted of (in mmol/L) 10 HEPES, 134 NaCl, 6 KCl, 2 $CaCl_2$, 10 glucose, and 1 $MgCl_2$ (adjusted to pH 7.4 with NaOH). The intracellular pipette solution for perforated-patch configuration consisted of (in mmol/L) 10 HEPES, 30 KCl, 10 NaCl, 110 K-aspartate, and 1 $MgCl_2$ (adjusted to pH 7.2 with NaOH). Cells were kept at room temperature in a bathing solution consisting of (in mmol/L) 10 HEPES, 134 NaCl, 6 KCl, 2 $CaCl_2$, 10 glucose, and 1 $MgCl_2$ (adjusted to pH 7.4 with NaOH). Narishige PC-100 puller (Narishige International USA, Inc, Amityville, NY) was utilized to pull patch electrodes, which were polished using MicroForge MF-830 polisher (Narishige International USA, Inc). The pipette resistance was (3–5 ΩM). Amphotericin B was dissolved in the intracellular pipette solution to reach a final concentration of 0.3 µmol/L. The data were acquired using HEKA EPC 10 amplifier and PatchMaster v2x90 program (Harvard Bioscience, Holliston, MA) and analyzed using FitMaster v2x73.2 (Harvard Bioscience) and MATLAB R2018a (MathWorks, Natick, MA). TRPV4 channel current was recorded from freshly isolated ECs as described previously (*Sonkusare et al., 2012*; *Ottolini et al., 2020b*). Briefly, GSK101-induced outward currents through TRPV4 channels were assessed in response to a 200 ms voltage step from –45 mV to +100 mV in the presence of ruthenium red in order to prevent $Ca^{2+}$ and activation of IK/SK channels at negative voltages.

TRPV4 channel current was recorded in HEK293 cells using whole-cell patch configuration 48 hr after transfection. The intracellular solution consisted of (in mmol/L) 20 CsCl, 100 Cs-aspartate, 1 $MgCl_2$, 4 ATP, 0.08 $CaCl_2$, 10 BAPTA, 10 HEPES, pH 7.2 (adjusted with CsOH). The extracellular solution

consisted of (in mmol/L) 10 HEPES, 134 NaCl, 6 KCl, 2 CaCl$_2$, 10 glucose, and 1 MgCl$_2$ (adjusted to pH 7.4 with NaOH). Currents were measured using a voltage clamp protocol where voltage-ramp pulses (–100 mV to +100 mV) were applied over 200 ms with a holding potential of –50 mV. TRPV4 currents were measured before or 5 min after treatment.

## Quantitative polymerase chain reaction (qPCR)

Mouse mesenteric arteries were denuded by pushing air through the arteries for 1 min. RNA was isolated using a Direct-zol RNA Miniprep (R2051, Zymo Research, Irvine, CA), with an in-column DNA Removal Kit. cDNA was converted with Bio-Rad iScript cDNA Synthesis Kit (1708841, Hercules, CA). The qPCR reaction mixes were prepared using Bio-Rad 2x  SsoAdvanced Universal SYBR Green Supermix (1725272, Hercules, CA), 200 nmol/L primers (Panx1_F: 5' TGCACAAGTTCTTCCCCTACA, Panx1_R: ATGGCGCGGTTGTAGACTTT; GAPDH_F: GGTTGTCTCCTGCGACTTCA; GAPDH_R TAGG GCCTCTCTTGCTCAGT; Eurofins Genomics Louisville, KY), and 20 nmol/L cDNA, then run in a Bio-Rad CFX96 qPCR Detection System. Results were analyzed using the $2^{-\Delta\Delta Ct}$ method.

## Statistical analysis

Results are presented as mean ± SEM. The n = 1 was defined as one artery in the imaging experiments (Ca$^{2+}$ imaging, PLA), one cell for patch-clamp experiments, one mouse for RVSP measurements, one artery for pressure myography experiments, one mouse for functional MRI, one mouse for ATP measurements, and one mouse for qPCR experiments. The data were obtained from at least three mice in experiments performed in at least two independent batches. The individual data points are shown for each dataset. For in vivo experiments, an independent team member performed random assignment of animals to groups and did not have knowledge of treatment assignment groups. All the in vivo experiments were blinded; information about the groups or treatments was withheld from the experimenter or from the team member who analyzed the data. All data are shown in graphical form using CorelDraw Graphics Suite X7 (Ottawa, ON, Canada) and statistically analyzed using GraphPad Prism 8.3.0 (Sand Diego, CA). A power analysis to determine group sizes and study power (>0.8) was performed using GLIMMPSE software ($\alpha$ = 0.05; >20% change). Using this method, we estimated at least  cells per group for patch-clamp experiments, five arteries per group for imaging and pressure myography experiments, and  mice per group for RVSP measurements and MRI. A Shapiro–Wilk test was performed to determine normality. The data in this article were normally distributed; therefore, parametric statistics were performed. Data were analyzed using two-tailed, paired or independent t-test (for comparison of data collected from two different treatments), one-way ANOVA or two-way ANOVA (to investigate statistical differences among more than two different treatments). Tukey correction was performed for multiple comparisons with one-way ANOVA, and Bonferroni correction was performed for multiple comparisons with two-way ANOVA. Statistical significance was determined as a p-value <0.05.

## Acknowledgements

The mouse strain *Cdh5*-optoα1AR was developed by CHROMus, which is supported by the National Heart Lung Blood Institute of the National Institute of Health under award number R24HL120847. This work was supported by grants from the National Institutes of Health to SKS (R01HL142808, R01HL146914, R01HL157407), BEI (P01HL120840, HL137112), and VEL (R01HL133293, R01HL157407).

## Additional information

### Funding

| Funder | Grant reference number | Author |
|---|---|---|
| National Institutes of Health | HL146914 | Swapnil K Sonkusare |
| National Institutes of Health | HL142808 | Swapnil K Sonkusare |

| Funder | Grant reference number | Author |
|---|---|---|
| National Institutes of Health | HL157407 | Victor E Laubach Swapnil K Sonkusare |
| National Institutes of Health | P01HL120840 | Brant E Isakson |
| National Institutes of Health | HL137112 | Brant E Isakson |
| National Institutes of Health | R01HL133293 | Victor E Laubach |

The funders had no role in study design, data collection and interpretation, or the decision to submit the work for publication.

## Author contributions

Zdravka Daneva, Data curation, Formal analysis, Investigation, Methodology, Validation, Writing – original draft, Writing – review and editing; Matteo Ottolini, Data curation, Formal analysis, Investigation, Methodology, Visualization; Yen Lin Chen, Data curation, Investigation, Methodology, Validation, Visualization; Eliska Klimentova, Investigation, Methodology, Validation; Maniselvan Kuppusamy, Data curation, Formal analysis, Investigation; Soham A Shah, Investigation, Methodology; Richard D Minshall, Cheikh I Seye, Brant E Isakson, Methodology, Resources; Victor E Laubach, Funding acquisition, Resources, Writing – review and editing; Swapnil K Sonkusare, Conceptualization, Data curation, Formal analysis, Funding acquisition, Investigation, Methodology, Project administration, Resources, Supervision, Validation, Visualization, Writing – original draft, Writing – review and editing

## Author ORCIDs

Zdravka Daneva (iD) http://orcid.org/0000-0002-1141-9697
Swapnil K Sonkusare (iD) http://orcid.org/0000-0001-9587-9342

## Ethics

All animal protocols were approved by the University of Virginia Animal Care and Use Committee (protocols 4100 and 4120). This study was performed in strict accordance with the recommendations in the Guide for the Care and Use of Laboratory Animals of the National Institutes of Health. For surgical procedures, every effort was made to minimize suffering.

## Decision letter and Author response

Decision letter https://doi.org/10.7554/67777.sa1
Author response https://doi.org/10.7554/67777.sa2

# Additional files

## Supplementary files
• Transparent reporting form

## Data availability

All data generated or analyzed during this study are included in the manuscript. Individual numeric values are shown in the scatterplots for each dataset. An Excel sheet with source data for Figure 1J has been provided.

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
