## [Decision Letter]

**Acceptance summary:**

This study, which makes a connection between several proteins known to regulate endothelial function in pulmonary arteries, may be of interest to vascular, pulmonary and ion channel physiologists. The study provides compelling evidence that ATP released from pulmonary artery endothelial cell pannexin1 channels activates TRPV4 channelsthat is facilitated by the scaffolding protein Caveolin-1 and that this pathway helps to maintain low pulmonary vascular resistance and pulmonary artery pressure. Identification of this pathway provides new drug targets to improve pulmonary endothelial function in disease states such characterized by impaired endothelial function.

**Decision letter after peer review:**

Thank you for submitting your article "Endothelial Pannexin 1-TRPV4 channel signaling lowers pulmonary arterial pressure" for consideration by *eLife*. Your article has been reviewed by 3 peer reviewers, one of whom is a member of our Board of Reviewing Editors, and the evaluation has been overseen by Richard Aldrich as the Senior Editor. The following individual involved in review of your submission has agreed to reveal their identity: William Jackson (Reviewer #3).

Essential revisions:

1. The major concern is related to conceptual significance. The reviewer appreciates that the work presented here connects Cav-1, Panx1, P2Y2R, PKC and TRPV4 into a signaling axis regulating pulmonary artery reactivity. However, this group has already published similar papers implicating a role for this axis in pulmonary arteries (and a similar axis in systemic arteries), and comparable conclusions have been reached by examining members of the pathway independently. Therefore, it is unclear what new conceptual information is gained, other than the link between all the proteins in the complex. Perhaps the authors could highlight more the major gaps in knowledge and novel aspects of their work. It should be clear to the reviewers what is novel and new.

2. A critical step in the proposed pathway is activation of Pannexin 1. The authors should provide evidence or at least clearly discuss how pannexin 1 could be activated to engage the proposed pathway.

3. The physiological role of the proposed pathway is unclear. PAP is normally low (8 – 20 mm Hg). Are the authors proposing that this pathway is always engaged to maintain low PAP? If so, then how is Pannexin 1 being tonically activated? This would also imply that there exists a tonic constrictor pathway which Pannexin1-TRPV4 opposes. Does this exist? Or the proposed Panx1-V4 pathway only engaged in the face of pulmonary hypertension. It is hard to envision a dilatory pathway when the system is already at low pressure, i.e., relaxed.

4. The use of the term, "small, resistance-sized pulmonary arteries" is curious. Pulmonary arteries have low resistance and pressure. What is the basis of using this term?

5. Previous work from this group has shown that endothelial ATP activates TRPV4 channels to cause nitric oxide dependent vasodilation of pulmonary arteries (PMID: 29275372 and PMID: 32463112). The current study has many of the same elements of previous studies. The authors should clearly what is new and what has been shown before.

6. Authors proposed that shear stress and flow can cause release of ATP in physiology condition and cited previous work from Yamamoto et al. (PMID: 12714321). It is important that authors perform some functional experiments to confirm this mechanism with TRPV4 and Panx1 KOs.

7. Authors' measurements show that approximately 400 nM ATP is released from PA endothelial cells, but they have used 1 µM for dilation experiments (Figure 1B) and 10 µM for all other experiments, which is 6-100-fold higher concentration than the measured concentration. Authors needs to provide justification for that.

8. Page 10: authors mentioned that PPADS and JNJ 47965567 did not inhibit TRPV4 sparkets, but Figure 2D shows significant inhibition which potentially suggest involvement of other purinergic (P2Xs) signaling.

9. Why did authors use different concentrations of activators such as GSK101 (1-30 nM) with different KO animals? Justification needed.

10. Figure 5E, authors suggested that PAP regulation is via NO signaling, but they did not provide any experimental evidence using eNOS inhibitors or eNOS KO animals.

11. Authors mentioned that they have used both Male and Female mice. Did they observe any differences between the sexes?

12. It is unclear why 1 nM GSK101 (Figure 1D) or 10 nM GSK101 (Figure 1G) induced reduced TRPV4 activity/current density in Panx1EC-/- compared to Panx1fl/fl. These observations do not support the statement on Page 6, Line 127-129, indicating that "reduced TRPV4EC channel activity in Panx1EC-/- is due to impaired channel regulation…". Since GSK101 is bypassing activation of the Panx1 pathway to stimulate TRPV4 channels directly, the expectation is that channel activity/current density should be similar provided that all other conditions are similar. A deeper consideration of this issue will help clarify concerns. The authors may consider performing immunofluorescence experiments to show that TRPV4-associated fluorescence or distribution of TRPV4 is similar in EC from Panx1fl/fl and Panx1EC-/-.

13. The statement on Page 6, Line 131-132 is not supported by the data presented. Figure 1 does not show the direct regulation of TRPV4 by Panx1. This statement should be revised. The U466 experiments in Figure 1I are interesting but do not link TRPV4 and Panx1. These experiments show that genetic ablation of any of these proteins increases pulmonary artery reactivity to U466. Can the increase in contraction to U466 in Panx1EC-/- reversed by treating the arteries with the TRPV4 agonist?

14. A major conclusion of the study is the formation of a nanocomplex between Cav-1, Panx1, P2Y2R, PKC and TRPV4. However, the interaction between these proteins is only presented for a subset of protein pairs. The study will be strengthened by providing data showing nanometer proximity between Panx1-P2Y2R, Panx1-TRPV4, P2Y2R-TRPV4 and that genetic ablation of Cav-1 disrupt the proximity between all these protein pairs. Based on the authors' data, P2Y1R should not activate TRPV4 channels. Thus, it will be expected that P2Y1R are not part of the complex. Is this the case? Please show antibody validation and negative controls for PLA. The authors should also explain why they think that Cav-1, Panx1, P2Y2R, PKC and TRPV4 are in nanometer proximity of each other given that no super-resolution data was presented.

15. Is ATP-induced dilation of small diameter pulmonary arteries prevented in Panx1EC-/-, P2Y2REC-/- and Cav-1EC-/-?

16. The description of Figure 2A gives the impression that perhaps all the ATP may come from EC. How much of the ATP will be left if the endothelial layer is removed before performing the assay? What are the ATP levels when using Panx1SMC-/- arteries? The study will be strengthened by linking mechanisms mediating basal ATP release with activation of the proposed pathway.

17. The sparklet traces shown in Figure 4D (recorded with X-Rhod in Cdh5-optoα1AR EC) is ascribed to TRPV4 channels, but there is no evidence that this is the case. This is important as data suggest that these sparklet events have a higher activity before light stimulation than other TRPV4 sparklet data presented throughout the manuscript. Are the amplitude and kinetics of sparklet events in this figure match that of known TRPV4 sparklets?

18. Is TRPV4 current density diminished in Cav-1EC-/-?

19. Authors should show evidence that TRPV4 and Panx1 expression is reduced/knockout in TRPV4SMC-/- and Panx1SMC-/-, respectively.

20. The immunofluorescent images in Figures 1A, 1C, 2C and 3A should be better described in the main text.

21. page 5, line 111 – I'm a bit puzzled why there was no right hypertrophy in your models with elevated PAP? Was it just that the duration of elevated PAP was insufficient to cause right heart hypertrophy? Please discuss.

22. page 6, lines 125-127 – Were currents different between control and EC Panx1-/- with 30 nM GSK101 in your patch clamp experiments? Why were higher concentrations of GSK101 used in the patch clamp experiments?

23. page 6, lines 129-131 – Did knock out of EC TRPV4-/- or EC Panx1-/- cause PAs to develop myogenic tone?

24. page 6 line 134 and onward – in your ex vivo ca^2+^ imaging experiments, what is the stimulus that is leading to Panx1 activity and release of ATP? It would seem important to identify the stimulus. Also, does luminal apyrase in pressure myograph experiments have the same effect as Panx1 knockout? How about P2Y2R-/- in pressure myography experiments? These experiments also would seem to be important to close the loop.

25. page 10, line 227 – What is the physiological stimulus for Panx1 and ATP release?

26. page 11, lines 236-239 – How do you reconcile the lack of effect of global TRPV4 knockout on PAP with your finding that EC TRPV4-/- increases PAP? Shoudln't global do the same thing? This should be better discussed.

27. page 16, lines 351-353 – You did not cannulate the pressure catheter – you cannulated the external jugular vein for access to the right ventricle – please revise to clarify.

28. Table 2 – How did you determine the specificity of the antibodies used?

29. Figure legends – please provide exact n-values for each panel and also please provide exact p-values for all statistical tests.

---

## [Author Response]

Essential revisions:1. The major concern is related to conceptual significance. The reviewer appreciates that the work presented here connects Cav-1, Panx1, P2Y2R, PKC and TRPV4 into a signaling axis regulating pulmonary artery reactivity. However, this group has already published similar papers implicating a role for this axis in pulmonary arteries (and a similar axis in systemic arteries), and comparable conclusions have been reached by examining members of the pathway independently. Therefore, it is unclear what new conceptual information is gained, other than the link between all the proteins in the complex. Perhaps the authors could highlight more the major gaps in knowledge and novel aspects of their work. It should be clear to the reviewers what is novel and new.

We thank the reviewers and the Editor for identifying the strengths of the manuscript and for their constructive feedback. We recently reported that endothelial TRPV4 channels decrease the contractility of small pulmonary arteries (PAs) and lower resting pulmonary arterial pressure (PAP)^1^. Moreover, exogenous ATP activated endothelial TRPV4 channels to dilate PAs^2^. However, the regulation of TRPV4 channel activity by endogenously released ATP, the source of endogenously released ATP, and the precise signaling mechanisms for ATP activation of endothelial TRPV4 channels were not known. In the current manuscript, we present a novel signaling axis whereby ATP efflux through endothelial Pannexin 1 (Panx1) activates nearby TRPV4 channels via purinergic receptor signaling to lower PA contractility and PAP. Following key findings contribute to the high conceptual significance and novelty of the study:

1) First evidence, using endothelial knockout mice, that ATP efflux through endothelial Panx1 lowers PA contractility and PAP. Notably, previous studies have shown that endothelial Panx1 activity does not contribute to vasodilation in systemic arteries and systemic blood pressure regulation^3^.

2) First direct evidence that ATP efflux through Panx1 promotes endothelial TRPV4 channel activity in PAs, but TRPV4 channel activity does not regulate ATP efflux through Panx1 under resting conditions.

3) First evidence that ATP effluxed through endothelial Panx1 stimulates purinergic P2Y2 receptor (P2Y2R) signaling to activate TRPV4 channels and lower PA contractility and resting PAP.

4) Earlier, we showed that endothelial caveolin-1 (Cav-1) lowers the resting PAP^1^. In the current manuscript, we provide evidence that endothelial Cav-1 provides a signaling scaffold for Panx1, P2Y2R, and TRPV4 channels, ensuring their spatial proximity in PAs. Activation of the endothelial Panx1–P2Y2 receptor–TRPV4 channel pathway, enabled by the Cav-1 scaffold, lowers PA contractility and PAP.

5) PAs are a high-flow vascular bed, yet flow-induced endothelial signaling is poorly understood in PAs. We provide evidence that physiological flow/shear stress increases luminal ATP release through endothelial Panx1 activation.

We have now modified the Introduction and other sections of the manuscript to highlight the conceptual significance and novelty of the results presented in this manuscript.

2. A critical step in the proposed pathway is activation of Pannexin 1. The authors should provide evidence or at least clearly discuss how pannexin 1 could be activated to engage the proposed pathway.

Using pressure myography in small PAs, we now show that endothelial Panx1-dependent signaling lowers pressure-induced (myogenic) constriction of PAs, thus exerting a dilatory effect under resting conditions. Regarding physiological activators of Panx1, flow/shear stress has been shown to activate ATP efflux through Panx1^4^. PAs are a high-flow vascular bed. However, flow/shear stress-activated signaling in PAs remains entirely unknown. In the revised manuscript, we provide evidence that flow/shear stress increases luminal ATP levels in PAs through endothelial Panx1 activation. Specifically, increased flow/shear stress elevated luminal ATP levels in pressurized PAs, an effect that was absent in PAs from endothelial Panx1^-/-^ mice (Figure 2G). Together, these findings support a dilatory effect of endothelial Panx1 under resting conditions and activation of endothelial Panx1 by flow/shear stress.

3. The physiological role of the proposed pathway is unclear. PAP is normally low (8 – 20 mm Hg). Are the authors proposing that this pathway is always engaged to maintain low PAP? If so, then how is Pannexin 1 being tonically activated? This would also imply that there exists a tonic constrictor pathway which Pannexin1-TRPV4 opposes. Does this exist? Or the proposed Panx1-V4 pathway only engaged in the face of pulmonary hypertension. It is hard to envision a dilatory pathway when the system is already at low pressure, i.e., relaxed.

We thank the reviewer for raising these important points. The reviewer has correctly noted that in this manuscript we propose a tonic endothelial Panx1-TRPV4 signaling axis that maintains a low PAP. It has generally been considered that PAs, due to low intraluminal pressure, are relaxed/low-resistance. However, this assumption results from the lack of studies on pressure-induced (myogenic) constriction in small PAs under resting conditions. In this manuscript we provide evidence that small PAs (50-100 microns) show myogenic constriction (~ 20% at 15 mm Hg), whereas large PAs (> 200 microns) do not (Figure 2B, 2D, 3G; Supplemental Figure 2A). Further, we show that PAs from endothelial Panx1^-/-^, TRPV4^-/-^, and P2Y2R^-/-^ mice develop significantly higher myogenic constriction compared to PAs from the respective control mice (Figure 2B, 2D and 3G). These data strongly support tonic activation of endothelial Panx1–P2Y2R–TRPV4 channel pathway and its dilatory effect under basal conditions. In addition to myogenic constriction, agonist-induced constriction was also higher in PAs from endothelial Panx1^-/-^, TRPV4^-/-^, and P2Y2R^-/-^ mice compared to the control mice (Figure 2C, 2E, and 3H). The detailed studies of myogenic constriction of PAs and mechanisms involved will be published in a separate manuscript.

As the reviewer pointed out, it is plausible that the Panx1-dependent signaling is altered in pulmonary hypertension, a possibility that has not been tested. In this regard we have shown that endothelial TRPV4 channel activity is impaired in PAs from PH patients and mouse models of PH^1^.

4. The use of the term, "small, resistance-sized pulmonary arteries" is curious. Pulmonary arteries have low resistance and pressure. What is the basis of using this term?

While the general opinion is that PAs are low-resistance or are in a completely relaxed state, there are no detailed studies showing a lack of myogenic constriction in pressurized small PAs under basal conditions. Our new PA pressure myography data show that small PAs (~ 50-100 microns) develop pressure-induced/myogenic constriction, whereas large PAs (~ 200 microns or more) do not (Figure 2B, 2D, and 3G, Supplemental Figure 2A). We use the term “resistance PAs” to describe PAs that show myogenic constriction (~ 50-100 microns, used in this study). We present evidence that PAs develop myogenic constriction at the physiological intraluminal pressure (15 mm Hg, Figure 2B, 2D, and 3G). We also show that PAs from endothelial Panx1^-/-^, TRPV4^-/-^, and P2Y2R^-/-^ mice develop significantly higher myogenic constriction compared to PAs from the respective control mice. Thus, endothelial knockout of Panx1, P2Y2R, or TRPV4 channel increases PA contractility and elevates PAP.

5. Previous work from this group has shown that endothelial ATP activates TRPV4 channels to cause nitric oxide dependent vasodilation of pulmonary arteries (PMID: 29275372 and PMID: 32463112). The current study has many of the same elements of previous studies. The authors should clearly what is new and what has been shown before.

We previously showed that exogenous ATP (1-10 µM) dilates PAs through endothelial TRPV4-eNOS signaling^1, 2^. However, the regulation of TRPV4 channel activity by endogenously released ATP, the source of endogenously released ATP, and the signaling mechanisms for ATP activation of endothelial TRPV4 channels were not known. The crucial new findings in this manuscript (not published elsewhere in the literature) include: (1) endothelial Panx1 is the predominant source of ATP efflux in PAs; (2) efflux of ATP through endothelial Panx1 enhances the activity of endothelial TRPV4 channels; (3) Panx1-effluxed extracellular ATP activates TRPV4 channels via endothelial P2Y2R–protein kinase C signaling; (4) Cav-1 provides a signaling scaffold for endothelial Panx1–P2Y2R–TRPV4 channel signaling; (5) endothelial Panx1–P2Y2R–TRPV4 channel pathway opposes myogenic and agonist-induced constriction of PAs and lowers PAP; and (6) flow/shear stress activates ATP release through endothelial Panx1 in PAs.

We have now revised the Introduction and other sections of the manuscript to highlight the novel findings.

6. Authors proposed that shear stress and flow can cause release of ATP in physiology condition and cited previous work from Yamamoto et al. (PMID: 12714321). It is important that authors perform some functional experiments to confirm this mechanism with TRPV4 and Panx1 KOs.

We performed a series of experiments to address this comment. We studied luminal ATP levels in pressurized PAs (15 mm Hg) in response to different flow/shear stress levels. Increased flow/shear stress elevated the luminal ATP levels in PAs from control mice, but this response was absent in PAs from endothelial Panx1^-/-^ mice (Figure 2G). Interestingly, flow/shear stress-induced increase in luminal ATP levels was not altered in PAs from endothelial TRPV4^-/-^ mice (Figure 2H), suggesting that TRPV4 channels are not involved in flow-induced activation of endothelial Panx1.

7. Authors' measurements show that approximately 400 nM ATP is released from PA endothelial cells, but they have used 1 µM for dilation experiments (Figure 1B) and 10 µM for all other experiments, which is 6-100-fold higher concentration than the measured concentration. Authors needs to provide justification for that.

Thank you for raising this important issue. In our ATP assays (Figure 1B), we measured the ATP levels in the solution containing PAs. We postulated that the concentration of local ATP generated close to TRPV4 channels may be higher than that recorded in the solution. We used 1 μmol/L ATP for PA pressure myography, 10 μmol/L for patch-clamp experiments, and 1 μmol/L ATP for ca^2+^ imaging (apologies for the typo in the initial submission). 1 μmol/L ATP was sufficient to get a dilation in PAs and an increase in endothelial TRPV4 sparklet activity. For patch-clamp experiments in isolated endothelial cells, some cells responded (increase in TRPV4 currents) to 1 μmol/L ATP whereas others did not. However, 100% of the cells responded to 10 μmol/L ATP in patch-clamp experiments. This could be due to the fact that patch-clamp experiments were performed at room temperature and in enzymatically isolated ECs, which could make them slightly less sensitive to extracellular ATP.

8. Page 10: authors mentioned that PPADS and JNJ 47965567 did not inhibit TRPV4 sparkets, but Figure 2D shows significant inhibition which potentially suggest involvement of other purinergic (P2Xs) signaling.

We performed further statistical analyses in response to this comment. Our data showed that TRPV4 sparklet activity in the presence of ATP was not different amongst control, +PPADS, and + JNJ groups, although the data appeared to be trending towards a decrease in activity with PPADS/JNJ. To explore this possibility further, we increased the n numbers for PPADS and JNJ (previously n=3, Figure 3B). However, the difference was still not significant amongst the groups (n=5). Therefore, we concluded that in PAs, ATP activates TRPV4 sparklets mainly through P2Y2R activation. Also, please note that in endothelial P2Y2 receptor knockout mice, ATP was not able to activate TRPV4 sparklets (Figure 3A). While these data do not completely rule out a role for P2X signaling in ATP-activation of TRPV4 channels in PAs, they do confirm an essential role for endothelial P2Y2R.

9. Why did authors use different concentrations of activators such as GSK101 (1-30 nM) with different KO animals? Justification needed.

TRPV4 sparklet measurements in the intact endothelium at 37°C are more sensitive to GSK101 than patch-clamp experiments in enzymatically isolated ECs at room temperature. Therefore, the concentration of GSK101 is lower in calcium imaging experiments than patch-clamp experiments. For example, 1 nmol/L GSK101 does not activate TRPV4 currents in patch-clamp experiments but shows considerable TRPV4 sparklet activity in the intact endothelium. Below is a detailed description of the concentrations of GSK101 used and the underlying reasons:

1. For determining the effect of a treatment on TRPV4 sparklet activity, we compared baseline activity (in the absence of GSK101) or that in the presence of a low concentration of GSK101 (1 nmol/L). This level of sparklet activation makes it easier to discern a change of activity with a treatment or genetic deletion.

2. The maximum TRPV4 sparklet activity in PAs is seen at 30 nmol/L GSK101^1^. Therefore, this concentration of GSK101 was used to determine whether the maximum number of functional channels is altered by a genetic deletion.

3. For patch-clamp experiments performed in isolated ECs at room temperature, we used 10 nmol/L GSK101 to consistently get TRPV4 currents without getting maximal channel activation. Next, we used 100 nmol/L GSK101 to obtain near-maximal activation of the channel^1^ and to rule out the effect of genetic deletion on maximum functional channels.

10. Figure 5E, authors suggested that PAP regulation is via NO signaling, but they did not provide any experimental evidence using eNOS inhibitors or eNOS KO animals.

We apologize for not citing our previous studies on eNOS-NO signaling downstream of TRPV4 channels. We previously reported that endothelial TRPV4 sparklets dilate PAs via eNOS activation. Specifically, TRPV4 channel activation increased NO levels, an effect that was absent in PAs from eNOS^-/-^ mice^2^. Moreover, TRPV4 channel-induced vasodilation was abolished by NOS inhibitor L-NNA. Also, in PAs from endothelial TRPV4^-/-^ mice, endothelial NO levels were reduced^1, 2^. We have now cited these studies in the revised manuscript (lines 336 – 339).

11. Authors mentioned that they have used both Male and Female mice. Did they observe any differences between the sexes?

Previous studies reported lower TRPV4 channel activity in cerebral pial and parenchymal myocytes from female mice than male mice^5^. We used male and female mice for RVSP measurements and vasodilation, and did not observe any sex differences. Therefore, we now specify in the Methods section that no sex differences were observed (lines 377-378).

12. It is unclear why 1 nM GSK101 (Figure 1D) or 10 nM GSK101 (Figure 1G) induced reduced TRPV4 activity/current density in Panx1EC-/- compared to Panx1fl/fl. These observations do not support the statement on Page 6, Line 127-129, indicating that "reduced TRPV4EC channel activity in Panx1EC-/- is due to impaired channel regulation…". Since GSK101 is bypassing activation of the Panx1 pathway to stimulate TRPV4 channels directly, the expectation is that channel activity/current density should be similar provided that all other conditions are similar. A deeper consideration of this issue will help clarify concerns. The authors may consider performing immunofluorescence experiments to show that TRPV4-associated fluorescence or distribution of TRPV4 is similar in EC from Panx1fl/fl and Panx1EC-/-.

Higher concentrations of GSK101 were used to get maximal channel activity in ECs from PAs (30 nmol/L for calcium imaging and 100 nmol/L for patch-clamp^1, 6^), indicating maximum number of functional channels. Our data show that the maximum number of functional channels are not different between endothelial Panx1^-/-^ and control mice (Supplemental Figure 1B). However, significant differences in activity were observed at lower levels of TRPV4 activation (1 nmol/L GSK101 for calcium imaging and 10 nmol/L for patch-clamp). Importantly, baseline (absence of GSK101) TRPV4 channel activity was also reduced in ECs from endothelial Panx1^-/-^ mice (Figure 1D). Thus, Panx1 deletion reduced TRPV4 channel activity but not maximal functional channels. We agree that these data could also be explained by channel mis-localization and are not a direct evidence for “impaired channel regulation”. Therefore, we have removed the statement on impaired channel regulation in the revised manuscript.

13. The statement on Page 6, Line 131-132 is not supported by the data presented. Figure 1 does not show the direct regulation of TRPV4 by Panx1. This statement should be revised. The U466 experiments in Figure 1I are interesting but do not link TRPV4 and Panx1. These experiments show that genetic ablation of any of these proteins increases pulmonary artery reactivity to U466. Can the increase in contraction to U466 in Panx1EC-/- reversed by treating the arteries with the TRPV4 agonist?

We performed additional myography experiments, where U46619-induced constriction in PAs from endothelial Panx1^-/-^ and P2Y2R^-/-^ mice was studied in the presence of a low concentration of GSK101 (3 nmol/L) (Figure 2E). GSK101 reduced U46619-induced constriction to the control levels seen in PAs from endothelial Panx1^fl/fl^ and P2Y2R^fl/fl^ mice. These data have been added to Figures 2E and 3H.

14. A major conclusion of the study is the formation of a nanocomplex between Cav-1, Panx1, P2Y2R, PKC and TRPV4. However, the interaction between these proteins is only presented for a subset of protein pairs. The study will be strengthened by providing data showing nanometer proximity between Panx1-P2Y2R, Panx1-TRPV4, P2Y2R-TRPV4 and that genetic ablation of Cav-1 disrupt the proximity between all these protein pairs. Based on the authors' data, P2Y1R should not activate TRPV4 channels. Thus, it will be expected that P2Y1R are not part of the complex. Is this the case? Please show antibody validation and negative controls for PLA. The authors should also explain why they think that Cav-1, Panx1, P2Y2R, PKC and TRPV4 are in nanometer proximity of each other given that no super-resolution data was presented.

PLA puncta indicate that two proteins are present within 40 nm of each other. In the initial version of this manuscript, we showed PLA data confirming the co-localization of endothelial Cav-1 with Panx1, P2Y2R, and TRPV4.

In response to the reviewer’s comment, we have now added PLA data demonstrating co-localization between TRPV4:P2Y2R and Panx1:P2Y2R (Figure 4E). Importantly, the PLA puncta were almost abolished when endothelial Cav-1 was genetically ablated, implying that endothelial Cav-1 is required for TRPV4:P2Y2R and Panx1:P2Y2R proximity. We also attempted PLA experiments for the one remaining protein pair- Panx1:TRPV4 using many different commercial antibodies. However, we could not draw meaningful conclusions due to non-specific signal with these new antibodies.

Our PLA experiments to detect potential co-localization between P2Y1R and Cav-1 (Supplemental Figure 4) demonstrated that P2Y1R does not co-localize with Cav-1. These data stand in agreement with the data that P2Y1R inhibition does not alter ATP activation of TRPV4 sparklets (Figure 1B).

The specificity of TRPV4, Panx1, P2Y2R, and Cav-1 antibodies was determined using PAs from the knockout mice as shown in Figure 1A, 3A, and Daneva et al., *PNAS*, 2021 ^1^. The specificity of the PKC antibody was tested using a competing peptide, as described earlier^7^. The negative controls for PLA experiments are now provided in the form of loss of PLA signal in endothelial Cav-1^-/-^ mice for the following pairs: (1) Cav1:TRPV4; (2) Cav-1:Panx1; and (3) Cav-1:P2Y2R. We have also provided negative controls for TRPV4:P2Y2R and Panx1:P2Y2R pairs as loss of signal in PAs from endothelial P2Y2R^-/-^ mice the PLA experiments (Supplemental Figure 4).

15. Is ATP-induced dilation of small diameter pulmonary arteries prevented in Panx1EC-/-, P2Y2REC-/- and Cav-1EC-/-?

We conducted additional pressure myography experiments to address this comment. ATP-induced dilation was reduced in PAs from endothelial TRPV4^-/-^, P2Y2R^-/-^, and Cav-1^-/-^ mice (Supplemental Figure 2C; Figure 3F and 4A). These data supported the idea that exogenous ATP dilates PAs through endothelial P2Y2R–TRPV4–signaling facilitated by Cav-1. ATP dilation, however, was not altered in PAs from endothelial Panx1^-/-^ mice (Supplemental Figure 2D), suggesting that Panx1 is upstream of ATP-P2Y2R-TRPV4 signaling.

16. The description of Figure 2A gives the impression that perhaps all the ATP may come from EC. How much of the ATP will be left if the endothelial layer is removed before performing the assay? What are the ATP levels when using Panx1SMC-/- arteries? The study will be strengthened by linking mechanisms mediating basal ATP release with activation of the proposed pathway.

In the new Figure 1B, we demonstrated reduced ATP release in PAs from endothelial Panx1^-/-^ mice compared to the control mice. We have now added data that endothelial denudation shows a similar decrease in ATP levels, suggesting that Panx1 is the predominant source of ATP release from ECs. Notably, the ATP levels in PAs from smooth muscle Panx1^-/-^ mice were significantly higher than those in PAs from endothelial Panx1^-/-^ mice, suggesting that the contribution of ECs to extracellular ATP is higher than that of SMCs. Finally, we show that ATP levels in endothelium-denuded PAs from smooth muscle Panx1^-/-^ mice are lower than endothelium-denuded PAs from control mice. These data confirmed that both EC and SMC Panx1 releases ATP, although ECs appear to be the predominant source of ATP under basal conditions.

17. The sparklet traces shown in Figure 4D (recorded with X-Rhod in Cdh5-optoα1AR EC) is ascribed to TRPV4 channels, but there is no evidence that this is the case. This is important as data suggest that these sparklet events have a higher activity before light stimulation than other TRPV4 sparklet data presented throughout the manuscript. Are the amplitude and kinetics of sparklet events in this figure match that of known TRPV4 sparklets?

In the initial submission we only presented the number of TRPV4 sparklet sites per cell with X-Rhod-1. NP_O_ could not be determined as the quantal level with X-Rhod-1 was not known. To address this concern from the reviewer, we first determined the quantal level of TRPV4 sparklets with X-Rhod-1 (0.21 ΔF/F0, presented in Supplemental Figure 5B). We are now able to present the X-Rhod-1 data as TRPV4 sparklet sites per field and sparklet activity (NP_O_) per site. Overall, initial TRPV4 sparklet activity was not different between fluo-4 and X-Rhod-1.

Further, in the presence of TRPV4 inhibitor GSK219 (100 nmol/L), light activation did not increase calcium signals (Supplemental Figure 5A), suggesting a specific effect on TRPV4 channels in PA endothelium.

18. Is TRPV4 current density diminished in Cav-1EC-/-?

We recently reported that pulmonary artery ECs from Cav-1_EC_^-/-^ mice showe reduced TRPV4 current density compared to the ECs from control mice^1^. We now cite this study and have added the statement reflecting a decrease in TRPV4 current density in Cav-1_EC_^-/-^ mice (lines 203-204).

19. Authors should show evidence that TRPV4 and Panx1 expression is reduced/knockout in TRPV4SMC-/- and Panx1SMC-/-, respectively.

We now provide evidence that Panx1 expression is reduced in our inducible, smooth muscle cell (SMC)-specific Panx1^-/-^ mice (Supplemental Figure 1A). We previously provided evidence that the expression of TRPV4 channel in SMCs is reduced in SMC-specific TRPV4^-/-^ mice^1^.

20. The immunofluorescent images in Figures 1A, 1C, 2C and 3A should be better described in the main text.

The immunofluorescent images in figures 1A and 3A have been described in more detail in the Results section (lines 108-109).

21. page 5, line 111 – I'm a bit puzzled why there was no right hypertrophy in your models with elevated PAP? Was it just that the duration of elevated PAP was insufficient to cause right heart hypertrophy? Please discuss.

The observation that there is no significant right ventricular hypertrophy in our genetic Panx1 and P2Y2R mouse models is in congruence with previous findings from our laboratory^1^. We agree with the reviewer that this could be due to a short duration of elevated PAP. All the experiments were performed two weeks after the last tamoxifen injection. While we get a PAP phenotype under these conditions, the duration of elevated PAP may not be sufficient to cause ventricular hypertrophy. This possibility has now been discussed (lines 286-290).

22. page 6, lines 125-127 – Were currents different between control and EC Panx1-/- with 30 nM GSK101 in your patch clamp experiments? Why were higher concentrations of GSK101 used in the patch clamp experiments?

We previously showed that 100 nmol/L GSK101 results in near-maximal activation of TRPV4 channel currents in pulmonary artery ECs^1^. We used two concentrations of GSK101 for our patch-clamp experiments: 10 nmol/L to get a low-level activation of TRPV4 channels; and 100 nmol/L to get near-maximal activation of TRPV4 channels^1^. 100 nmol/L GSK101 was used to determine the number of functional channels. We postulated that at lower levels of activity (10 nmol/L), the difference in TRPV4 currents due to genetic ablation of Panx1/P2Y2R will be more discernible. At a high concentration (100 nmol/L), the direct channel agonist will surpass the channel regulation by Panx1–ATP–P2Y2R signaling and result in maximal channel activation regardless of the absence of the regulatory proteins. Consistent with this, we previously showed that genetic deletion of endothelial Cav-1 results in reduced TRPV4 currents at 10 nmol/L GSK101 but not at 100 nmol/L GSK101^1^.

23. page 6, lines 129-131 – Did knock out of EC TRPV4-/- or EC Panx1-/- cause PAs to develop myogenic tone?

We performed PA pressure myography experiments to carefully test if endothelial knockout of Panx1, TRPV4, or P2Y2R results in increased myogenic constriction at 15 mm Hg. We show that PAs from endothelial Panx1, TRPV4, and P2Y2R knockout mice show higher myogenic constriction compared to PAs from the respective control mice (Figure 2B, 2D, and 3G). In combination with the data on U46619-induced constriction, these results support the concept that endothelial Panx1–P2Y2R–Cav-1–TRPV4 signaling axis lowers myogenic and agonist-induced constriction of PAs.

24. page 6 line 134 and onward – in your ex vivo ca^2+^ imaging experiments, what is the stimulus that is leading to Panx1 activity and release of ATP? It would seem important to identify the stimulus. Also, does luminal apyrase in pressure myograph experiments have the same effect as Panx1 knockout? How about P2Y2R-/- in pressure myography experiments? These experiments also would seem to be important to close the loop.

Our ATP measurements in isolated PAs (Figure 1B) show that ATP is being effluxed from EC Panx1 under resting/baseline conditions. We postulate that there is a similar basal ATP efflux through Panx1 in our imaging experiments, although the exact stimulus is not known. It is possible that Panx1 is activated by intracellular ca^2+^ or changes in membrane potential or membrane stretch. It is important to note that TRPV4 channels do not appear to be altering ATP efflux through Panx1 under basal conditions (Figure 1B).

Regarding physiological activators of Panx1, flow/shear stress has been shown to activate ATP efflux through Panx1^4^. PAs are a high-flow vascular bed. However, flow/shear stress-activated signaling in PAs remains entirely unknown. In the revised manuscript, we provide evidence that flow/shear stress increases luminal ATP levels in PAs through endothelial Panx1 activation. Specifically, increased flow/shear stress elevated luminal ATP levels in pressurized PAs, and this effect was absent in PAs from endothelial Panx1^-/-^ mice (Figure 2G). The shear stress is negligible in our calcium experiments (< 0.01 dynes/cm^2^). It is not clear if this level of shear stress can activate Panx1.

Furthermore, we conducted pressure myography experiments where PAs were treated with luminal apyrase (10 U/mL), followed by U46619. We observed that apyrase pretreatment increased the contractility of PAs from control mice (Supplemental Figure 2B) to the level of PA from endothelial Panx1^-/-^ mice in absence of apyrase (Figure 2E), further supporting the dilatory effect of ATP effluxed through Panx1.

We have also added new experiments in PAs from endothelial P2Y2R^-/-^ mice. First, we show that the myogenic constriction is increased in PAs from endothelial P2Y2R^-/-^ mice compared to the control mice (Figure 3G). Next, we demonstrate that constriction to U46619 is significantly higher in PAs from endothelial P2Y2R^-/-^ mice compared to the control mice (Figure 3H). These data are in agreement with the results that RVSP is elevated in endothelial P2Y2R^-/-^ mice compared to the control mice (Figure 3E).

25. page 11, lines 236-239 – How do you reconcile the lack of effect of global TRPV4 knockout on PAP with your finding that EC TRPV4-/- increases PAP? Shoudln't global do the same thing? This should be better discussed.

Cell-specific knockout mice offer a significant advantage over global knockout mice with respect to discerning cell-type specific phenotype. For an endothelium-specific effect, the phenotype is expected to be better in an inducible, endothelium-specific knockout mouse than a global knockout mouse. Indeed, global TRPV4 knockout mice showed no systemic blood pressure or pulmonary arterial pressure phenotype^8, 9^. However, inducible, endothelium-specific knockout mice had elevated systemic blood pressure and pulmonary arterial pressure^1^. Lack of a phenotype in global knockout mice could be due to the deletion of TRPV4 channels from multiple cell types or compensatory mechanisms that have developed over time. These issues with the use of non-inducible, global knockout mice are well-documented in the literature. These possibilities have now been discussed (lines 275-281).

26. page 16, lines 351-353 – You did not cannulate the pressure catheter – you cannulated the external jugular vein for access to the right ventricle – please revise to clarify.

We thank the reviewer for this correction. This mistake has now been fixed (line 407-408).

27. Table 2 – How did you determine the specificity of the antibodies used?

The specificity of TRPV4, Panx1, P2Y2R, and Cav-1 antibodies was determined using PAs from the knockout mice as shown in Figure 1A, 3A, and Daneva et al., *PNAS*, 2021^1^. The specificity of the PKC antibody was tested using a competing peptide, as described earlier^7^. The negative controls for PLA experiments are now provided in the form of loss of PLA signal in endothelial Cav-1^-/-^ mice for the following pairs: (1) Cav1:TRPV4; (2) Cav-1:Panx1; and (3) Cav-1:P2Y2R. We have also provided negative controls for TRPV4:P2Y2R and Panx1:P2Y2R pairs as loss of signal in PAs from endothelial P2Y2R^-/-^ mice the PLA experiments (Supplemental Figure 4).

28. Figure legends – please provide exact n-values for each panel and also please provide exact p-values for all statistical tests.

We have now added n-values and p-values for each panel in the figure legends.

1. Daneva Z, Marziano C, Ottolini M, Chen YL, Baker TM, Kuppusamy M, Zhang A, Ta HQ, Reagan CE, Mihalek AD, Kasetti RB, Shen Y, Isakson BE, Minshall RD, Zode GS, Goncharova EA, Laubach VE and Sonkusare SK. Caveolar peroxynitrite formation impairs endothelial TRPV4 channels and elevates pulmonary arterial pressure in pulmonary hypertension. Proceedings of the National Academy of Sciences of the United States of America. 2021;118.

2. Marziano C, Hong K, Cope EL, Kotlikoff MI, Isakson BE and Sonkusare SK. Nitric Oxide-Dependent Feedback Loop Regulates Transient Receptor Potential Vanilloid 4 (TRPV4) Channel Cooperativity and Endothelial Function in Small Pulmonary Arteries. *J Am Heart Assoc*. 2017;6.

3. Good ME, Eucker SA, Li J, Bacon HM, Lang SM, Butcher JT, Johnson TJ, Gaykema RP, Patel MK, Zuo Z and Isakson BE. Endothelial cell Pannexin1 modulates severity of ischemic stroke by regulating cerebral inflammation and myogenic tone. *JCI Insight*. 2018;3.

4. Wang S, Chennupati R, Kaur H, Iring A, Wettschureck N and Offermanns S. Endothelial cation channel PIEZO1 controls blood pressure by mediating flow-induced ATP release. *J Clin Invest*. 2016;126:4527-4536.

5. Tajada S, Moreno CM, O'Dwyer S, Woods S, Sato D, Navedo MF and Santana LF. Distance constraints on activation of TRPV4 channels by AKAP150-bound PKCalpha in arterial myocytes. *J Gen Physiol*. 2017;149:639-659.

6. Sonkusare SK, Bonev AD, Ledoux J, Liedtke W, Kotlikoff MI, Heppner TJ, Hill-Eubanks DC and Nelson MT. Elementary Ca2+ signals through endothelial TRPV4 channels regulate vascular function. *Science*. 2012;336:597-601.

7. Ottolini M, Hong K, Cope EL, Daneva Z, DeLalio LJ, Sokolowski JD, Marziano C, Nguyen NY, Altschmied J, Haendeler J, Johnstone SR, Kalani MY, Park MS, Patel RP, Liedtke W, Isakson BE and Sonkusare SK. Local Peroxynitrite Impairs Endothelial TRPV4 Channels and Elevates Blood Pressure in Obesity. *Circulation*. 2020.

8. Xia Y, Fu Z, Hu J, Huang C, Paudel O, Cai S, Liedtke W and Sham JS. TRPV4 channel contributes to serotonin-induced pulmonary vasoconstriction and the enhanced vascular reactivity in chronic hypoxic pulmonary hypertension. *Am J Physiol Cell Physiol*. 2013;305:C704-15.

9. Yang XR, Lin AH, Hughes JM, Flavahan NA, Cao YN, Liedtke W and Sham JS. Upregulation of osmo-mechanosensitive TRPV4 channel facilitates chronic hypoxia-induced myogenic tone and pulmonary hypertension. *Am J Physiol Lung Cell Mol Physiol*. 2012;302:L555-68.